# TO THE CUTOFF... AND BEYOND? A LONGITUDINAL PERSPECTIVE ON LLM DATA CONTAMINATION

**Manley Roberts[1], Himanshu Thakur[1,2], Christine Herlihy[3], Colin White[1], Samuel Dooley[1]**
[1]Abacus.AI   [2]Carnegie Mellon University   [3]University of Maryland
{manley,colin,samuel}@abacus.ai; hthakur@andrew.cmu.edu;
cherlihy@umd.edu

## ABSTRACT

Recent claims about the impressive abilities of large language models (LLMs) are often supported by evaluating publicly available benchmarks. Since LLMs train on wide swaths of the internet, this practice raises concerns of data contamination, i.e., evaluating on examples that are intentionally or unintentionally included in the training data. Data contamination remains notoriously challenging to measure and mitigate, even with partial attempts like controlled experimentation of training data, canary strings, or embedding similarities. In this work, we conduct the first thorough longitudinal analysis of data contamination in LLMs by using the natural experiment of training cutoffs in GPT models to look at benchmarks released over time. Specifically, we consider two code/mathematical problem-solving datasets, Codeforces and Project Euler, and we find statistically significant trends among LLM pass rate vs. GitHub popularity and release date that provide strong evidence of contamination. By open-sourcing our dataset, raw results, and evaluation framework, our work paves the way for rigorous analyses of data contamination in modern models. We conclude with a discussion of best practices and future steps for publicly releasing benchmark in the age of LLMs which train on webscale data.

## 1 INTRODUCTION

Progress in machine learning has historically been driven by the use of benchmark datasets (Raji et al., 2021) to demonstrate and ultimately improve model performance. In recent years, as large language models (LLMs) have risen to prominence, these benchmarks are used to claim impressive capabilities across a wide range of tasks (Brown et al., 2020a), such as open-ended text and code generation. However, it has become increasingly clear that evaluating on these benchmarks jeopardizes our ability to accurately compare and assess modern models since static, open-source benchmarks are generally published on the internet, and most modern LLMs incorporate internet text in their training data.

There are two main phenomena to be concerned with. The first is *contamination*, which refers to an LLM's exposure, during training, to examples that are similar or identical to the examples that the model will later be evaluated on. The second is *memorization*, which can be understood as a property of a model that permits extraction of generated outputs that are exact or near-exact replicas of examples seen during training. Both phenomena can pose security and privacy risks (Carlini et al., 2021). Additionally, as we discuss below, they can upwardly bias model performance estimates, obfuscating our ability to compare models and attribute performance gains to true model improvements.

Despite these concerns, contamination and memorization remain deceptively challenging to *definitively* measure and detect. While some researchers have used string-matching algorithms to compare test to training datasets (Radford et al., 2019; Brown et al., 2020b), many popular LLMs' full training dataset details are not publicly available (OpenAI, 2023a; Rozière et al., 2023). Additionally, string-matching produces false negatives when slight variations exist in data between train and test (OpenAI, 2023a). Even concerted efforts to prevent any model from training on a benchmark can fail. For example, the canary strings present in all BIG-Bench files (bench authors, 2023), which are designed to be checked and excluded by model trainers, were not sufficient to keep BIG-bench out of GPT-4's training corpus (OpenAI, 2023a), partly because the success of this strategy relies on the awareness and compliance of model trainers in the absence of an enforcement mechanism.

Recent works that look for contamination or memorization focus on popular benchmarks. They use controlled experimentation on models trained with certain subsets of chosen datasets, recognizing the value of comparing performance on examples that are seen vs. not seen during training (Magar & Schwartz, 2022; Zhang et al., 2021).

In contrast, we take an experimental economics view and use a naturally occurring experiment—i.e., the training cut-off date—to assess contamination and memorization. We exploit the known training cutoff dates of GPT-4 and GPT-3.5-Turbo (OpenAI, 2023a;b) and assumed cutoff date of Code Bison (Google, 2023) to *naturally* partition benchmark examples into subsets that have either probably been seen (pre-cutoff) or have probably[1] not been seen (post-cutoff). We focus our analysis on longitudinal benchmarks consisting of problems released over a period of time which bridges the cutoff.

In particular, we analyze Codeforces and Project Euler, two longitudinal code generation/problem solving websites. These websites have steadily released problems since 2010 and 2001, respectively. Informal analyses have shown that there are large drops in success rates of GPT-4 when evaluated on older versus more recent problems from Codeforces (He, 2023; Cundy, 2023).

We build upon these insights by conducting the first rigorous, large-scale, longitudinal analysis of contamination and memorization in code generation and problem-solving benchmarks. To the best of our knowledge, we are the first to exploit the longitudinal nature of the benchmarks we analyze, along with the known training cutoff dates of the open and closed sourced models, to naturally identify examples that the LLMs are likely/unlikely to have been exposed to during training, and use this partition to compare LLM performance during the pre- and post-cutoff periods.

**Our contributions** In this work, we explore contamination and memorization through the lens of time. Our core contributions include:

(i) The first large-scale, longitudinal analysis of contamination and memorization using a naturally occurring experiment — a novel methodology in LLM contamination which is important in light of closed-source models;

(ii) Empirical findings demonstrating that GPT-4 was likely exposed to Codeforces and Project Euler, due to a statistically significant positive association we observe between a problem's presence on GitHub and each LLM's test case pass rate *only* for problems released before the GPT training cutoff;

(iii) Code required to construct our longitudinal datasets and perform analyses, which we open-source.[2]

## 2 RELATED WORK

**Evaluation of Code Generation Models**   Code generation models are generative models that try to produce valid code given an input of some representation of the programmatic behavior, mathematical function, and/or computational task that the user would like to obtain. Modern code generation models include general models such as GPT family (OpenAI, 2023a), Llama 2 (Rozière et al., 2023), or PaLM (Chowdhery et al., 2022), as well as a variety of task-specific code models: AlphaCode (Li et al., 2022), CodeGen (Nijkamp et al., 2022), Code-Llama (Rozière et al., 2023), PaLM-Coder (Chowdhery et al., 2022).Relevant code generation benchmarks include small sets of entirely handwritten problems (Chen et al., 2021; Nijkamp et al., 2022) as well as larger collections curated from internet sources such as code interview sites, competitive programming forums, or general open source code (Hendrycks et al., 2021; Austin et al., 2021; Zan et al., 2022; Huang et al., 2022), and some that include both original and online-sourced problems (Yin et al., 2022; Li et al., 2022). Code interview, practice, or competition sites, offering problem descriptions and programmatic evaluation, are common choices to assess modern LLM capabilities (Nguyen & Nadi, 2022; Zhang et al., 2023; He, 2023; Cundy, 2023)—and indeed some public benchmarks feature these problems (Hendrycks et al., 2021; Li et al., 2022).

---

[1]GPT-4 acknowledges training with some small amount of data beyond its cutoff (OpenAI, 2023a), so post-cutoff examples may still appear. GPT-3.5-Turbo, subject to similar reinforcement learning with human feedback (RLHF) as GPT-4 (OpenAI, 2023a), may have seen data beyond its cutoff as well.

[2]Our treatment of datasets and our evaluation framework are available at `https://github.com/abacusai/to-the-cutoff`. We release code and dataset contents to the extent possible while respecting the licensing requirements of the dataset owners.

To asses the validity of solutions, many of these benchmarks include test cases. They use a 'functional correctness' metric based on passing these cases as the primary way to measure code generation performance; evaluating with complexity/understandability metrics (Nguyen & Nadi, 2022) is less common. Kulal et al. (2019); Chen et al. (2021) employ the *pass@k* metric, describing the likelihood at least one among $k$ sampled generations will pass all test cases. The benefit of these metrics is the complete independence from either expensive human feedback or inherently constraining similarity-to-ground-truth NLP metrics (Papineni et al., 2002; Lin, 2004), which are often ineffective for code (Tran et al., 2019). These metrics are in contrast to other popular LLM performance metrics like perplexity (Kirchenbauer et al., 2023; Jain et al., 2023) or information retrieval based LLM metrics of accuracy (Kwiatkowski et al., 2019; Pal et al., 2023).

**Adversarial Filtering and Adaptive Benchmarks in NLP**   Test-time exploitation of knowledge gained via contamination or memorization can be seen as special cases of a more general phenomenon in which language models *appear* to exhibit sophisticated reasoning capabilities but are in fact exploiting shallower heuristics, with potentially negative consequences for generalizability (Bender et al., 2021). Prior work has demonstrated that domain-agnostic and domain-specific crowd worker-constructed natural language inference (NLI) datasets—i.e., SNLI (Bowman et al., 2015), MultiNLI (Williams et al., 2018), MedNLI (Romanov & Shivade, 2018)—contain spurious correlations between lexical and syntactic features of the inputs and the corresponding class labels, such that hypothesis-only baselines (i.e., without premise) are able to outperform majority-class baselines (Poliak et al., 2018; Gururangan et al., 2018; McCoy et al., 2019; Herlihy & Rudinger, 2021). Researchers have proposed a variety of detection and mitigation strategies, including *(1)* adversarial filtering, in which an ensemble of classifiers are used to iteratively partition a dataset into *easy* and *hard* subsets (Zellers et al., 2018); *(2)* introduction of stochasticity to the annotator prompting process via randomly selected anchor words (Sakaguchi et al., 2020); and *(3)* calls for the development of *adversarially adaptive* rather than static benchmarks (Zellers et al., 2019).

**Memorization and Contamination in LLMs**   Many recent works have highlighted the security, privacy, and generalizability risks of memorization and contamination during LLM training and fine-tuning, while simultaneously proposing methods for detection and risk mitigation. Mireshghallah et al. (2022); Biderman et al. (2023); Carlini et al. (2023); Magar & Schwartz (2022) investigate the training dynamics of memorization/contamination, seeking scaling laws, early indications, and understanding of when and how memorization occurs in training. Carlini et al. (2021) famously extract hundreds of verbatim train examples from GPT-2. Ippolito et al. (2023) propose inference-time tricks to prevent regurgitation of examples, and Jacovi et al. (2023); Karmakar et al. (2022) give best practices to avoid benchmark contamination. Carlini et al. (2021; 2023); Lee et al. (2022); Kandpal et al. (2022); Magar & Schwartz (2022); Carlini et al. (2019) investigate the relationship between duplicated training data and memorization/contamination (in particular, Carlini et al. (2019) uses artificially introduced "canary" artifacts to track memorization). Nori et al. (2023) proposes a distance-based metrics to assess memorization. Several works (Magar & Schwartz, 2022; Zhang et al., 2021) evaluate the impact of memorization/contamination by estimating the difference in test-time performance on examples seen vs. not seen during training; we will use a variation of this strategy.

Dodge et al. (2021) conduct a case study of the webcrawl corpus C4, including contamination investigation, while others (Aiyappa et al., 2023; Chang et al., 2023; Golchin & Surdeanu, 2023) conduct studies on the contamination of GPT models directly. Karmakar et al. (2022) dive deep into Hackerrank (an interview-prep coding platform) contamination in the Codex model by not only assessing pass rates on full problems but also on partial problem snippets. Golchin & Surdeanu (2023), a recent work, focuses in particular on comparing the result of prompting for memorized completion with or without benchmark clues and concludes that GPT-4 has been contaminated with several standard datasets; our analysis finds more contamination of GPT-4, but differs by examining *longitudinal* datasets in order to view the effects of dataset portions before and after training cutoffs.

## 3   DATASET CONSTRUCTION

Many open-source benchmarks  (Chen et al., 2021) designed to evaluate code generation are released at a certain point in time, evaluated on a number of models along with release, and then deployed repeatedly as time goes on in order to evaluate new models' performance on the benchmark. For

a model with a strict temporal training dataset cutoff, these benchmarks exist either firmly within or outside of the training data, meaning that to evaluate the effect of the cutoff, we must compare between *multiple* datasets (which, clearly, might have many differences beyond their release dates).

For this analysis, we concern ourselves with datasets with hand-written original problems that are released at intervals over a long stretch of time. In particular, we require that a substantial number of problems are produced before and after the GPT-4/GPT-3.5-Turbo cutoffs in September 2021, that the bulk of problems are of a format and size sufficient for prompting to modern LLMs, and that there exists an automated objective measure of correctness for evaluation. We focus on problems from the competitive programming website Codeforces (problems from 2010 - 2023) (Mirzayanov, 2023) and from the mathematical programming puzzle website Project Euler (problems from 2001-2023) (Hughes, 2023), building off analyses from (Cundy, 2023; He, 2023).

**Codeforces** Codeforces is a website that hosts competitive programming competitions. Problems are released in small batches corresponding to a particular round, and competitors submit solutions against test cases, competing to produce the highest overall score by giving fast solutions. After a competition ends, each competitor's solutions are available online, as well as the test cases that were evaluated on each problem (which take the form of an input file and expected output).

For each problem, we collect metadata, problem text (processed to clear some HTML artifacts), and input/expected output text for public and private test cases. We forgo the compute-intensive procedure of generating additional test cases for problems which was used by (Li et al., 2022) and omit test cases in which either the given input or output on the Codeforces platform end with "...", as this *often* indicates that the text is too long and has been abridged. We provide additional details of the Codeforces problem set and our collection process in Appendix A.2.

**Project Euler** Project Euler is a website that hosts difficult math problems with a string answer that is usually a single number (integral or real). The recommended way to solve these problems is to write code that will generate the answer. The answer itself can be submitted on the site and compared directly to the private solution (there are no official public solutions). There are no test cases except a comparison with the true answer.

We collect Project Euler problems through a combination of their metadata API and direct scraping of problem pages. We collect problems through 845 (released May 2023) and use open-source solutions from (luckytoilet, 2023). These solutions were collected in September 2023, but there are a few recent problems through 845 without a solution from this source; these we omit.

## 4 Methodological Approach

The primary research questions we endeavor to explore through longitudinal analysis of pre- versus post-cutoff LLM performance include: *(1)* Does there exist a statistically significant relationship between a programming problem's frequency of presence in open-source GitHub repositories and an LLM's ability to generate a functionally correct solution to that problem, and/or reproduce portions of its metadata, such as the problem title or tags? *(2)* How or to what extent is this relationship mediated by a problem's reported difficulty? *(3)* Most critically—how or to what extent do (1) and (2) change depending on whether a problem was released *before* versus *after* the LLM's training date cutoff?

**Models** To answer these questions, we conduct analysis on output produced by *GPT-4*, *GPT-3.5.-Turbo*, *Davinci-002*, Google's *code-bison*, and Meta's *Code-Llama*. The specific models used are `gpt-4-0314`, `gpt-3.5-turbo-0301`, `text-davinci-002`, `code-bison@001`, and `codellama/CodeLlama-34b-Instruct-hf`.

**Independent Variables** To begin, we define the following set of independent variables (IVs):

*GitHub Presence* is a proxy metric intended to capture the frequency with which a problem is publicly available on GitHub (similar to public Google and Bing API search used by Chang et al. (2023) as a proxy for online presence of books). For simplicity, it searches only for mention of the problem's name and ID. To compute GitHub Presence, we begin by collecting all public repositories that contain mentions of the benchmark dataset of interest (i.e., Codeforces or Project

Euler) as of our collection date. Then, for each problem of interest in a given dataset, we filter the dataset repositories and retain the subset containing substring(s) that correspond to the problem's title. We are then able to approximately compute the number of times a problem $p$ occurs as: $\sum_{i=1}^{|\text{dataset repos}|} c(p, i) \; \forall p \in \{\text{dataset problems}\}$, where $c(p, i)$ is the number of matches within repo $i$'s concatenated text to any one of a number of format variations of $p$'s ID or title. Counting multiple occurrences within the same repo offers benefits such as a more granular analysis in the event of mega-repos that might store multiple solutions to the same problem, and it is therefore in our eyes a closer proxy to the true frequency of the problem in the training data.

*Difficulty* intuitively captures how challenging a problem is for humans to solve. Both Codeforces and Project Euler report difficulty scores as part of problem metadata.

*Problem released post-cutoff* is a Boolean variable to indicate whether a given problem was released (i.e., published by the dataset owners) *before* (0) or *after* (1) the training date cutoff for a given LLM.

**Dependent Variables** We consider the following set of dependent variables (DVs):

*Problem-level pass rate (pass rate)* We assume that in the general case, a given problem $p$ can be mapped to some number, $n_p \geq 1$ of test cases (either public or private). For code generation tasks, the question-level pass rate can then be computed as the fraction of test cases for which the code generated by the LLM produces a functionally correct solution—i.e., $\frac{1}{n_p} \sum_{i=1}^{n_p} \mathbb{1}(\lambda(LLM(p)) = y_i)$, where $\lambda$ represents calling the LLM's generated code and $y_i$ represents the ground-truth output for problem $p$'s $i^{\text{th}}$ test case. The special case where we ask the LLM to generate (only) the solution rather than code can be represented by omitting the $\lambda$ call in the above expression. We use code generation on Codeforces and solution-only generation on Project Euler. See Appendix A.1 for a discussion of alternative metrics.

*Title reproduction* In each of the datasets we consider, each problem has a title. To compute title reproduction for a given problem $p$, we provide as input the dataset name and problem ID, ask the LLM to generate the problem's title given this input, and evaluate the similarity between the generated string, $\hat{\text{title}}_p$, and $p$'s ground-truth title by mapping the title into a bag of tokens and modeling the retrieval of each token as a separate observation in logistic regression. We include this DV as a probe for possible memorization.

*Tag reproduction* Among the datasets we consider, only Codeforces problems contain descriptive tags. Each tag is an n-gram that describes the intended approach or content of a problem, as well as metadata like the difficulty. For example, Problem 500A has tags "dfs and similar", "graphs", "implementation", and "*1000". For a given problem, $p$, we provide the problem's title and ID as input to the LLM, and ask it to produce a set of candidate tags. We evaluate token-level recall with respect to the tokenized version of the problem's ground-truth tag(s). Much like title reproduction, this DV is included as a memorization probe.

To answer the aforementioned research questions for each dataset and dependent variable, we conduct regression analyses with problem-level performance as our unit of analysis, of the form:

$$\text{DV} \sim (\text{Difficulty} + \text{GitHub Presence}) \cdot \text{postCutoff}$$

Because the problem-level pass rate prediction task involves count data, we specifically formalize it as a binomial regression, such that for a given problem, $p$ with a corresponding number of {public + private} test cases, $n_p$, we seek to predict the number of successes—i.e., the number of test cases, out of $n_p$ trials that the LLM's generated code and/or numeric solution will pass. In title reproduction, the outcome of interest is binary—i.e., the LLM either *does* or *does not* successfully reproduce the problem's title; as such, we model this task using logistic regression. For tag reproduction, while a problem's tags can be set-valued, we tokenize the string of tags and evaluate the recall of each token independently; as such, this task is also modeled using logistic regression. A more detailed description of our modeling choices, along with interpretation guidance for the regression tables and marginal effects plots, can be found in Appendix B.1.

In the regression tables, we report coefficients as odds ratios where values equal to 1 indicate no impact of the variable on the Pass Rate. Coefficients greater than 1 indicate a positive impact and those less than 1 indicate a negative impact. For example, an odds ratio coefficient of 1.352 would correspond to a 35.2% increase in the dependent variable associated with a unit increase in the independent variable.

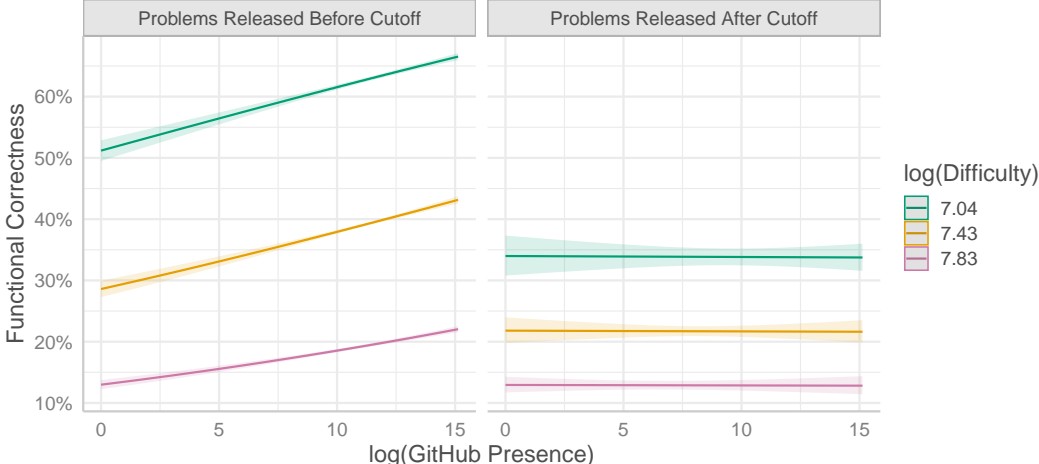

Figure 1: Marginal Effects of Pass Rate Metric for GPT-4 on the Codeforces Dataset. Observe a positive association between `GitHub Presence` before the cutoff but not after. Also, there is a negative association between `Difficulty` and pass rate both before and after the cutoff.

## 5 RESULTS

Overall, we see strong trends that the performance of each model changes after the training cutoff. These changes often highlight that there is a positive association between the presence of questions on GitHub and the performance of the model; however, after the training cutoff, this association disappears. We provide examples of the generations of the LLMs in Appendix B.8 for a qualitative inspection of the results. We note that, while we did test the code generation performance of the open source models *text-davinci-002* and *codellama/CodeLlama-34b-Instruct-hf*, these models' functional correctness performance was too low to yield meaningful analysis. Thus, we omit these models from all analyses in the main paper, but refer the reader to Appendix B.6.

### 5.1 PASS RATE

**GitHub Presence**    First, we look at the performance of Pass Rate on the benchmark Codeforces, where we report marginal effect plots for GPT-4 in Figure 1; GPT-3.5-Turbo and Code Bison are qualitatively similar and can be found in Appendix Figures 12 and 14. We report regression coefficients for all models on Codeforces in Figure 2. On the Project Euler benchmark, we report marginal effect plots in Appendix Figures 28 and 30 and regression coefficients in Figure 3. Note that Project Euler is a much smaller benchmark with just 73 problems included after the GPT training cutoff date in September 2021. Additionally, none of the LLMs we tested got any of the questions correct for this set of 73 problems. We make several observations.

Most strikingly, we see that the effect of the GitHub Presence variable is significant before the training cut-off and is not significant after the training cutoff. For GPT-4, we observe that for each increase in one unit of the log of GitHub Presence, we see the odds ratio increase by 4.5% on Codeforces and 47.8% on Project Euler; for GPT-3.5-Turbo, that value is moderated slightly to 2.5% on Codeforces and 27.7% on Project Euler; for Code Bison we see the odds ratio increase by 3.1%. However, we see no statistically significant association between GitHub Presence and GPT model performance for those problems which appeared online *after* the training cutoff in September 2021.

This post-cutoff performance degradation provides evidence of contamination and/or memorization of pre-cutoff problems from Codeforces and Project Euler by GPT-3.5-Turbo and GPT-4. For the most part, the odds ratios are similar in terms of the direction and magnitude of their effects on the pass rate odds for each LLM. Two points of distinction include: (1) GPT-4 performs better across the board, as evidenced by higher odds of functional correctness for all difficulty levels in both the pre- and post-cutoff periods as compared to GPT-3.5-Turbo. (2) For Codeforces, the odds ratio for GitHub

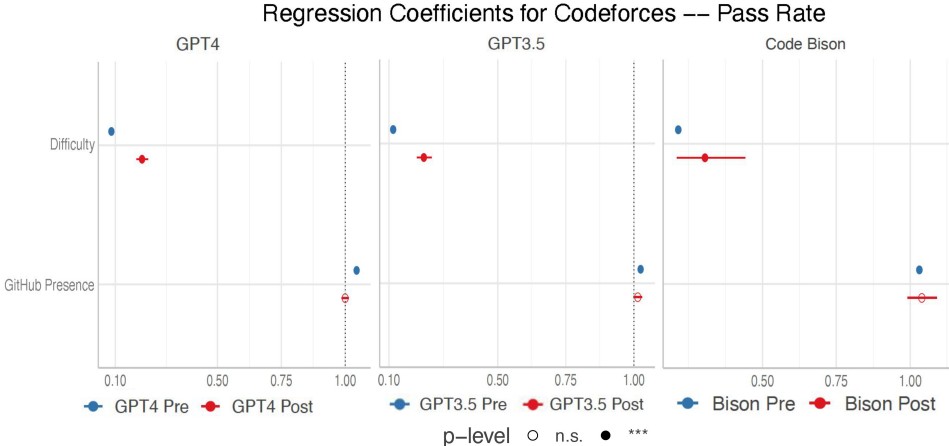

Figure 2: Regression coefficients for Pass Rate of GPT4, GPT-3.5-Turbo, and Code Bison on the Codeforces dataset. Observe that the odds ratios for both `Difficulty` and `GitHub Presence` are statistically significantly moderated between the before and after cutoffs for both models. See Table 1 and 2 for regression coefficients.

presence is equal to 1 and is not statistically significant during the post-cutoff period for GPT-4, but is $> 1$ (i.e., associated with increased odds of $Y$) and statistically significant for $\alpha = 0.1$ during the same period for GPT-3.5-Turbo (see Table 1 and 2). While training details for GPT-family models are generally secret, we propose as a possible explanation that GPT-3.5-Turbo may have had higher train/finetune exposure to problems released after the cutoff date than GPT-4[3].

It is worth pointing out that the relationships between Github Presence and pass rate have non-overlapping confidence intervals (before and after cutoff) for only GPT-4. This lends itself to our conclusion that memorization occurred for this model.

Code Bison's analysis uses a different cutoff (February 2023). On all of the data before this cutoff, there is a positive association between Github Presence and pass rate, while after there is no association. This analysis does produce a very large confidence band on the post-cutoff problems (due to the small sample size of Codeforces problems collected between Feb. 2023 and June 2023), making definitive conclusions difficult to resolve.

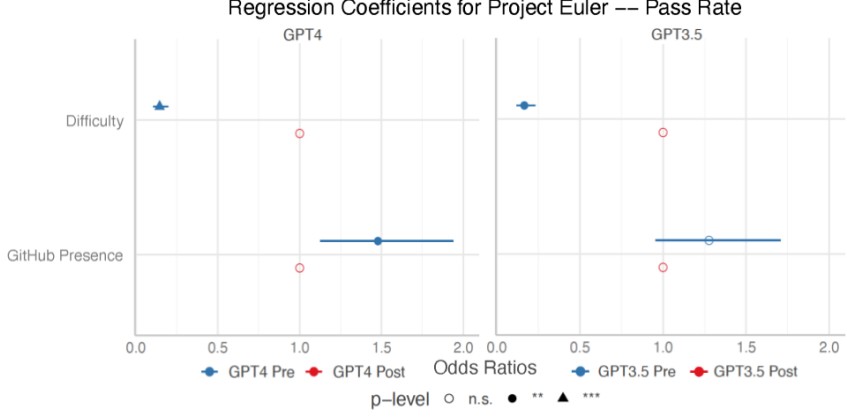

Figure 3: Regression coefficients plots of Pass Rate for GPT-4 and GPT-3.5-Turbo on the Project Euler Dataset. See Table 10 and 11 for regression coefficients. No problems pass after the cutoff.

---

[3]As mentioned in Section 1, GPT-4 is known to have some post-cutoff events included in its training; since GPT-3.5-Turbo uses a similar RLHF procedure (OpenAI, 2023a), it's possible it has been exposed as well—to a publicly unknown extent.

**Difficulty** When we examine results for GPT-4, GPT-3.5-Turbo, and Code Bison on Codeforces (see Tables 1, 2, 3), we see that there always exists a statistically significant, negative association between `Difficulty` and pass rate for each LLM—i.e., this relationship is observed in both the pre- *and* post-cutoff periods. However, while each model's post-cutoff `Difficulty` coefficient is $< 1$, indicating a decrease in the odds of pass rate these coefficients are statistically significantly larger than their corresponding pre-cutoff values, suggesting a moderation in the still-negative relationship between `Difficulty` and pass rate.

On the one hand, we can interpret the persistence of this relationship as evidence that the LLMs' inductive biases, while perhaps influenced by the effects of contamination and memorization, are by no means solely determined by such artifacts. For this reason, we do not see equal (or perhaps, equally poor) performance across problem difficulty levels in the post-period, but instead see that LLM pass rates vary in accordance with difficulty even in the (hypothetical) absence of contamination, much as they do for human programmers.

Other possible contributing factors include: (1) variation in the number of test cases by difficulty level, and/or over time; (2) more limited, but non-zero amounts of contamination or memorization of the datasets we analyze; and (3) the presence of unobserved confounder(s) influencing change in both problem difficulty and LLM pass rate over time. We test (1) by fitting a regression model to examine whether `Difficulty` is able to predict the number of observed test cases after the cutoff, but do not find `Difficulty` to have predictive power. Hypothesis (2) could be occurring, particularly given the acknowledged GPT fine-tuning (OpenAI, 2023a); however, it is unlikely this is happening at high enough levels to be a sufficient cause of observed behavior. We view the identification of possible confounders as a promising direction for future work.

## 5.2 Title and Tag Reproduction

For title reproduction, we show regression tables in Appendix Tables 12-15 and Appendix Figures 32-38. We conclude that across all models, there is no impact of GitHub Presence on the ability of the LLMs to reproduce the title, both before and after the training cutoffs.

For tag reproduction, we find that there is a negative association between GitHub Presence and the ability of the LLMs to reproduce the tag labels on Codeforces (there are no tags associated with Project Euler). In Figure 42, Appendix Figure 40 and Appendix Tables 16 and 17, we can see that across the board, there is a negative association between Difficulty and tag reproduction performance before the cutoff but there is no association after the cutoff. As the regression results demonstrate, the negative association moderates after the cutoff, dropping from a decrease of 56.9% to 17.4% in odds ratios from before to after the cutoff for GPT-4 and from 50.3% to 26.1% for GPT-3.5-Turbo.

The way in which Codeforces problems are available online is one hypothesis as to why tags reproduction is inversely related to GitHub presence, whereas title reproduction is not. Tags are metadata for Codeforces problems which are *not* present in the main problem description. As such, the tags may be less likely to be copied and pasted throughout the internet. Thus, it is possible that those tags themselves undergo some interesting distribution shift which could explain their *inverse* relationship with presence on GitHub.

## 5.3 Analysis Ablations

**Public vs Private Test Cases** As discussed in Section 3, the Codeforces problems contain both public and private test cases. Public cases are readily available on the problem's page whereas the private cases can only be found by opening an accepted submission on the Codeforces platform. Above, we analyzed the pass rate of each problem on all collected test cases. Now, we break these out by public and private test cases to investigate any different trends between the two sets. We consider only the private test cases in Figures 22-24 and Tables 7-8, whereas only the public test cases in Figures 16-18 and Tables 4-5.

We see first that the two main trends we observed above hold, indicating the robustness of the conclusions: contamination is likely since `GitHub Presence` is positively correlated with pass rate only before the cutoff, and `Difficulty` has a negative association with pass rate. However, we observe non-overlapping confidence intervals only for GPT-4 private test cases.

However, we also observe, unexpectedly, that the pass rate after the cutoff is higher for the private test cases than for the public test cases. This observation contrasts the typical perspective on Codeforces which considers the public test cases to be simpler toy cases used as examples while coding whereas the private cases are more thorough checks for correct behavior. To explain the discrepancy, we hypothesize that this behavior may be related to the *private* test cases after the cutoff being, on average, *easier to answer* than public test cases after the cutoff. There is no per-case difficulty score available on Codeforces, but we can consider a simple heuristic: shorter inputs are simpler to answer, and longer inputs are harder. Why might this effect be most noticeable *after the cutoff?* To answer, we observe that while the median test case input string lengths for our public and private pre-cutoff test cases are similar, at 18 and 21 characters, respectively, the median input lengths after the cutoff diverge for public and private test cases: 38 for public and 27 for private. Further investigation into the causes and consequences of this shift is a promising direction for future work.

**Covariate Shift**    We detail how we assess whether the performance degradation that we observe for problems released after the training cutoff might be caused by covariate shifts in the questions present in Codeforces and Project Euler. More precisely, we examine the distribution over tags and/or difficulty level, and we look for statistically significant changes in their prevalence during the post-cutoff period, relative to the pre-cutoff period. We visually inspect the distribution over tags (for Codeforces) and over discretized difficulty scores (for both datasets) for problems released during the pre- vs. post- periods, and do not find evidence of qualitative differences. We then conduct $\chi^2$ tests using the pre-cutoff normalized counts as the reference distribution. We do not find any statistically significant difference in any of the pre- versus post-distributions that we analyze. Plots and detailed statistical results are available in Appendix B.7.

# 6    DISCUSSION

**Utility of longitudinal analysis**: We provide a novel methodology for examining data contamination in LLMs, borrowed from experimental economics where we observe phenomena by exploiting naturally occurring changes. Thus, we present a novel way to approximately validate claims made about training date cutoffs for black box LLMs, and/or exposure (or lack thereof) to a given dataset during training or fine-tuning, provided that the dataset in question contains instances on each side of the model's reported cutoff. This can be valuable in cases where specific training details are not public, and/or when contamination or memorization is suspected as a root cause of performance degradation when the model in question is evaluated on newer problems. It is important to acknowledge that limitations also exist—for example, we cannot rule out the presence of latent confounder(s) influencing both exposure (i.e., to a given subset of problems) *and* LLM performance on those problems.

**Implications for LLM evaluation**: Our findings in Section 5 illustrate the extent to which even high-quality, manually constructed benchmarks can be expected to enjoy ever-shorter shelf lives in the era of LLMs, as newer models with updated training cutoff dates will iteratively render existing benchmarks stale. Detection of memorization and contamination will likely remain challenging in the general case, as many popular benchmarks have been released all at once rather than over time, and as such, cannot be subjected to longitudinal analyses like the ones we perform. Additionally, in open-ended domains such as code generation, we may fail to detect instances where the model has been exposed to solution(s) for a given problem when the problem context itself is missing or latent (i.e., by training on public repositories where people may not reproduce the questions their code is intended to answer/solve).

Current mitigation options, including the use of private (i.e., offline, closed-source) benchmarks, and/or benchmarks known to be constructed after the training cutoff for evaluation target(s) of interest, are likely to be time-bound in their utility and may be cost-prohibitive to sustain over longer time horizons, in the absence of exogenous shocks to the pace of LLM release cycles. Additionally, reliance on private benchmarks may further erode transparency and lead to duplication of efforts as challenging examples are detected and addressed in a more siloed fashion. Thus, while the need for some set of open-source "goalposts" against which to measure progress, evaluate, and compare LLMs is likely to persist, the way in which we construct, release, and evaluate against benchmark datasets will need to become more dynamic. We urge the community to move away from static benchmarks released in a single time step and toward continuous integration-style staggered release and evaluation cycles.

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

# A ADDITIONAL METHODOLOGY

## A.1 METRICS

We note that *Pass@k* (Chen et al., 2021) is an alternative outcome metric that is commonly reported within the LLM evaluation literature, which explores the number of sampled generations that pass every unit test of a problem. We can map our analyses onto Pass@1 since we generate one sample from each LLM. However, we also look at the problem pass rate (defined above) since the number of unit tests for Codeforces can be very large (sometimes over 500) and Pass Rate provides a more granular view of model performance than Pass@1. In this paper, Pass@1 is defined as the number of problems that pass all unit tests.

## A.2 CODEFORCES DATASET COLLECTION

(Caballero et al., 2016; Hendrycks et al., 2021; Li et al., 2022) assembled, in their respective datasets, a portion of Codeforces problems and test cases through 2018 and 2021, respectively. However, only (Li et al., 2022) contained any problems after the GPT-4/3.5-Turbo cutoff, and it had so few problems after the cutoff that we needed to collect Codeforces problems ourselves in order to have enough post-cutoff problems. In fact, we do not use their version of the problems at all, instead collecting a full set of problems from 2010-2023 ourselves, in order to ensure that the data quality is consistent for all problems. We do replicate many design decisions from Li et al. (2022), including separating the "public" test cases (those simpler test cases available in the problem description, fed to the model, and given during the competition to all competitors) from the "private" test cases (those test cases reserved in secret at the release of a competition but made public after some time).

We collected every problem on the Codeforces site through Codeforces Contest 1840, also known as round 843 (which took place June 6, 2023), and removed problems for a variety of reasons including: no test cases are found; testing is interactive and requires communicating with the Codeforces server; a problem has no English-language version; a problem is given as a PDF and not HTML text; or a problem is a special contest such as a private contest, a Q# quantum programming competition, or an April Fools' competition.

# B ADDITIONAL EMPIRICAL RESULTS

## B.1 REGRESSION DESCRIPTIONS

As opposed to other forms of regression, logistic regression uses odds rather than probabilities, and the main quantity explored is an odds ratio. For an event with probability $p$, the odds of that event is defined as $p/(1-p)$. Odds ratios explain the relationship between independent variables (predictors or features) and the probability of the binary outcome. In logistic regression, we use odds ratios to quantify the effect of a one-unit change in an independent variable on the odds of the binary outcome. The odds ratio is defined as the ratio of the odds of a problem passing to the odds that the problem doesn't pass.

More formally, let $Y$ be the binary outcome variable indicating failure/success of a question by an LLM where $Y \in \{0, 1\}$ and we assume $P(Y = 1) = p$. Let $X_1, \ldots, X_n$ be a set of predictor variables. Then the logistic regression of $Y$ on $X_1, \ldots, X_n$ estimates parameters $\beta_0, \ldots, \beta_n$ through maximum likelihood via the equation:

$$logit(p) = \ln(\frac{p}{1-p}) = \beta_0 + \beta_1 X_1 + \cdots + \beta_n X_n.$$

When we fit the regression, we obtain estimates for the $\beta_i$s. These fitted $\hat{\beta}_i$ can be interpreted as coefficients to the regression equation and provide intuition for how the independent variables $X_i$ influence the independent variable. Specifically, a fitted value of $\hat{\beta}_i$ tells us that, keeping all other variables constant, a one unit change in the variable $X_i$ yields a change of $\hat{\beta}_i$ in the log odds ratio of the independent variable. It is also common to exponentiate the fitted coefficients, in which case a unit change in $X_i$ while holding the other dependent variables constant yields a $e^{\hat{\beta}_i}$ change in the odds ratio of the independent variable.

These $e^{\hat{\beta}_i}$ are odds ratios that take values between 0 and $\infty$. They provide insight into how a change in the predictor variable $X_i$ affects the odds of the event occurring. If $e^{\hat{\beta}_i} = 1$, it suggests that the predictor variable has *no effect* on the odds of $Y$. If $e^{\hat{\beta}_i} > 1$, it suggests that an increase in $X_i$ is associated with *increased odds* of the event happening, specifically by providing a $(1 - e^{\hat{\beta}_i})\%$ increase in the odds of $Y$. If $e^{\hat{\beta}_i} > 1$, it suggests that an increase in $X_i$ is associated with *decreased odds* of the event happening, specifically by providing a $(1 - e^{\hat{\beta}_i})\%$ decrease in the odds of $Y$.

In our analyses, we are primarily interested in the independent variables `GitHub Presence`, `Difficulty`, and an indicator variable indicating whether a problem was released *before* or *after* the training cutoff. Below, we report regression tables with estimated odds ratio coefficients as well as marginal effects plots which visually depict the fitted regressions.

## B.2 PASS RATE

In this section, we present the marginal effects on Pass Rate of GPT-4 and GPT-3.5-Turbo across the Codeforces and Project Euler benchmarks.

### B.2.1 ALL CODEFORCES DATA

First, we present the marginal effects in Figures 10, 12, 14 and regression coefficients in Tables 1, 2, 3 with corresponding regression coefficient visualization in Figures 9, 11, 13.

### B.2.2 PUBLIC CODEFORCES DATA

Second, we evaluate only on the public test cases of Codeforces and produce the marginal effect plots in Figures 16, 18, 20 and regression coefficients in Tables 4, 5, 6 with corresponding regression coefficient visualization in Figures 15, 17, 19.

### B.2.3 PRIVATE CODEFORCES DATA

Finally, we consider only the private test cases and get the marginal effect plots shown in Figures 22, 24, 26 and regression coefficients in Tables 7, 8, 9 with corresponding regression coefficient visualization in Figures 21, 23, 25.

## B.3 PROJECT EULER

Finally, we report pass rates on Project Euler in Figures 28 and 30. We also present the regression coefficients in Tables 10 and 11 with corresponding regression coefficient visualization in Figures 27, 29.

## B.4 TITLE REPRODUCTION

Here, we present the marginal effects on the Title Reproduction metric for GPT-4 and GPT-3.5-Turbo across the Codeforces and Project Euler benchmarks in Figures 34, 32, 36 and 38. We also present the regression coefficients in Tables 12, 13, 14 and 15 with corresponding regression coefficient visualization in Figures 31, 33, 35, and 37.

## B.5 TAG REPRODUCTION

Here, we present the marginal effects on the Tag Reproduction metric for GPT-4 and GPT-3.5-Turbo across the Codeforces and Project Euler benchmarks in Figures 42 and 40. We also present the regression coefficients in Tables 17 and 16 with corresponding regression coefficient visualization in Figures 39, 41.

## B.6 EXPERIMENTS WITH ADDITIONAL LLMS

The results in Sections 5 and B.2-B.5 exclusively use the models GPT-3.5 and GPT-4. In this section, we discuss results on Davinci-002 and Code-Llama. Overall, our motivation to focus on GPT-3.5

and GPT-4 is because they are the only models to achieve nontrivial pass rate. For example, on Codeforces, GPT-3.5 achieved 27% and 13% pass rates before and after the cutoff, and GPT-4 achieved 37% and 16% pass rates before and after the cutoff, but the pass rates for text-davinci-002 with a comparable prompting strategy are both less than 1%. Similarly, a partial analysis of *codellama/CodeLlama-34b-Instruct-hf* featuring 3400+ randomly chosen problems from Codeforces (2800+ pre-cutoff, 500+ post-cutoff) yielded less than 1% pass rate both before and after the cutoff. Due to the very small number of problems that succeeded even before the cutoff date, we were unable to present non-trivial differences before and after the cutoff. On the other hand, we still include all raw output data from our experiments with text-davinci-002 in our supplementary material at `https://anonymous.4open.science/r/to-the-cutoff-review-253A`. We will shortly make our Code-Llama data available as well.

## B.7 ANALYSIS OF POSSIBLE COVARIATE SHIFTS

In this section, we conduct analyses to assess whether the drop-off in performance that we observe for problem examples released after the GPT training cutoff might be attributable to (potentially latent) covariate shifts. We specifically investigate whether the distribution over tags (only available for Codeforces) and/or difficulty level (available for Codeforces and Project Euler) changed in a statistically significant way during the post-cutoff period, relative to the pre-cutoff period. We summarize our findings by dataset below:

### B.7.1 CODEFORCES

**Tags**: In Codeforces, each problem is mapped by the problem creators to one or more descriptive tags. These tags generally reflect algorithm(s), data structure(s), and/or problem-solving approaches associated with the problem in question. We partition the Codeforces problems into pre- versus post (reflecting whether a given problem was released *before* or *after* the GPT training cutoff) and compute aggregate counts over tags, then normalize. We visualize the resulting frequencies in Figure 4 and observe that the two distributions are qualitatively quite similar:

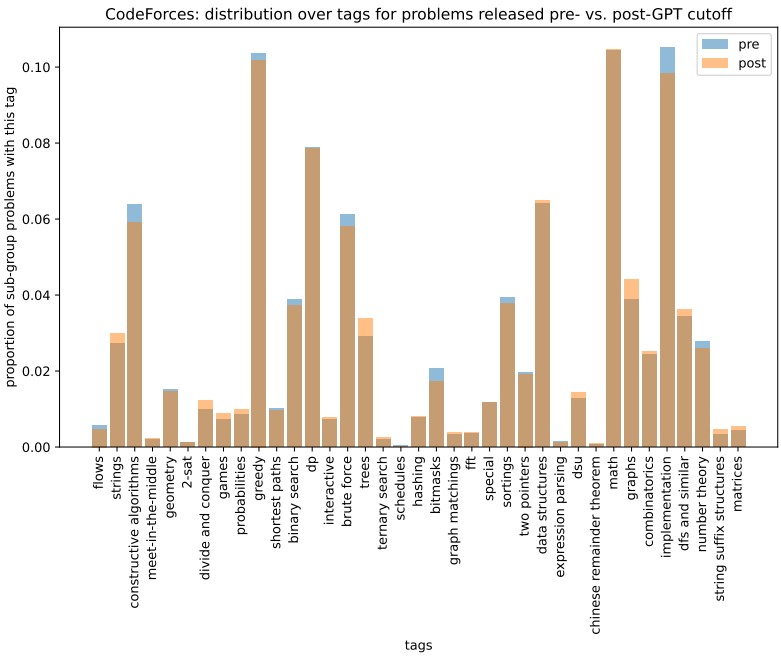

Figure 4: Codeforces: distribution over tags for problems released pre- vs. post-GPT cutoff

We also conduct a $\chi^2$ test to assess whether we are able to reject the null hypothesis, which states that there is no significant difference between the normalized distribution of counts over labels in the pre-versus post-cutoff examples (we use the pre-cutoff normalized counts as the reference or expected

distribution). We omit two tags with observed counts < 5; these are `chinese remainder theorem` and `schedules`. The resulting $\chi^2$ test statistic is 0.006 (p=1.0); thus, we are *not* able to reject the null hypothesis. These findings mitigate concerns that the drop-off in performance we observe for Codeforces might be attributable to significant changes in the distribution over tags during the post-cutoff period.

**Difficulty**: For the Codeforces problems we consider, difficulty takes values in the range [0,3500], with higher values corresponding to increased difficulty. We visualize the distribution over raw difficulty scores for problems in the pre-cutoff versus post-cutoff subsets in Figure 5:

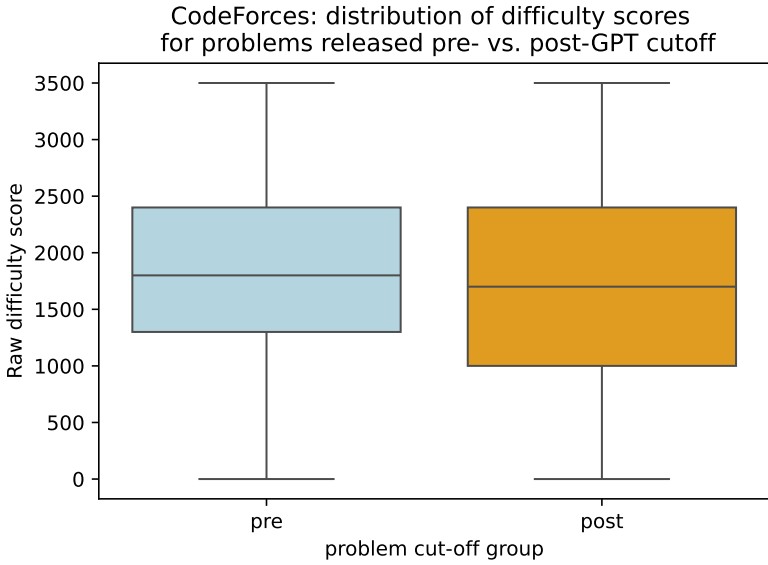

Figure 5: Codeforces: distribution over raw difficulty scores for problems released pre- vs. post-GPT cutoff

We then discretize the raw difficulty scores by mapping raw scores to discrete labels as follows:

$$\lambda(x) = \begin{cases} 0 & x \leq Q1 \\ 1 & Q1 < x \leq Q2 \\ 2 & Q2 < x \leq Q3 \\ 3 & x > Q3 \end{cases} \tag{1}$$

where $x$ represents a given problem's raw difficulty score, and Q1, Q2, and Q3 correspond to the first, second, and third quartiles, respectively. We visualize the pre-vs.post-cutoff problems' distribution over discretized difficulty scores in Figure 6:

We also conduct a $\chi^2$ test to assess whether we are able to reject the null hypothesis, which states that there is no significant difference between the normalized distribution of counts over discretized difficulty scores in the pre- versus post-cutoff examples (we use the pre-cutoff normalized counts as the reference or expected distribution). The resulting $\chi^2$ test statistic is 0.073 (p=0.995); thus, we are *not* able to reject the null hypothesis. These findings mitigate concerns that the drop-off in performance we observe for Codeforces might be attributable to significant changes in the distribution over difficulty levels during the post-cutoff period.

### B.7.2 PROJECT EULER

Note that as mentioned above, Project Euler does not make their problem tags publicly available; as such, we are not able to conduct tag analysis for this dataset, and we restrict our attention to difficulty.

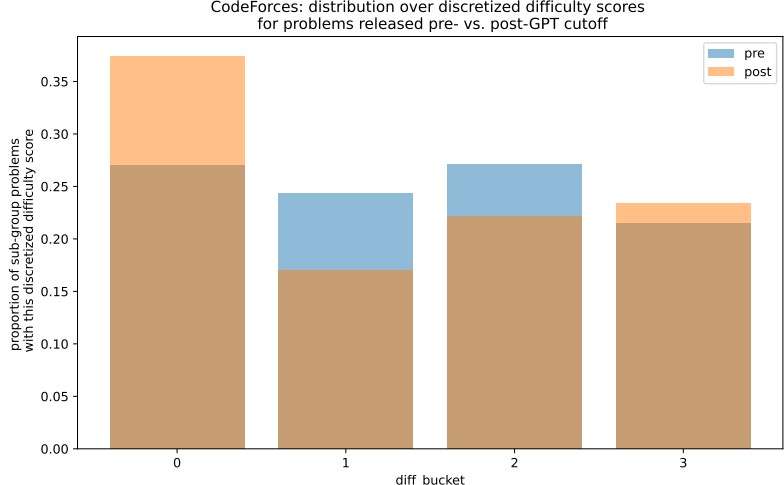

Figure 6: Codeforces: distribution over discretized difficulty scores for problems released pre- vs. post-GPT cutoff

**Difficulty**: For the Project Euler problems we consider, difficulty takes values in the range [5,100], with higher values corresponding to increased difficulty. We visualize the distribution over raw difficulty scores for problems in the pre-cutoff versus post-cutoff subsets in Figure 7:

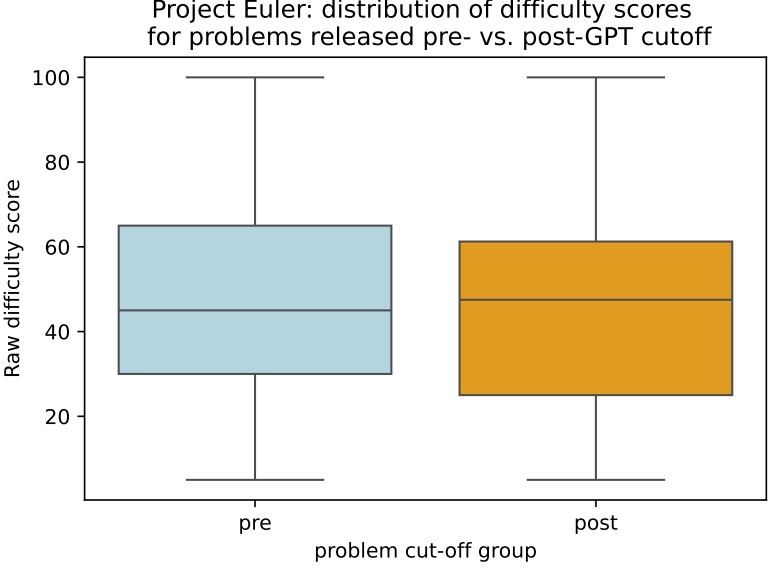

Figure 7: Project Euler: distribution over raw difficulty scores for problems released pre- vs. post-GPT cutoff

Similarly to our Codeforces difficulty analysis, we discretize the raw difficulty scores using the mapping in Equation 1. We visualize the pre-vs.post-cutoff problems' distribution over discretized difficulty scores in Figure 8:

We also conduct a $\chi^2$ test to assess whether we are able to reject the null hypothesis, which states that there is no significant difference between the normalized distribution of counts over discretized difficulty scores in the pre- versus post-cutoff examples (we use the pre-cutoff normalized counts as the reference or expected distribution). The resulting $\chi^2$ test statistic is 0.027 (p=0.999); thus, we are *not* able to reject the null hypothesis. These findings mitigate concerns that the drop-off

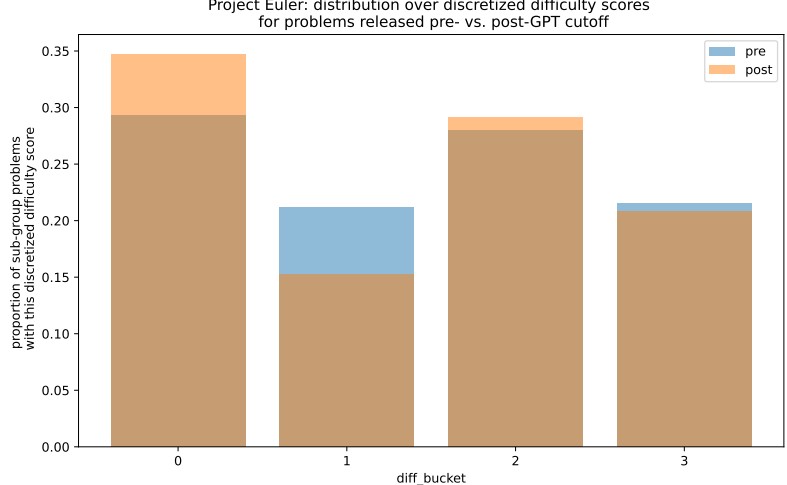

Figure 8: Project Euler: distribution over discretized difficulty scores for problems released pre- vs. post-GPT cutoff

in performance we observe for Project Euler might be attributable to significant changes in the distribution over difficulty levels during the post-cutoff period.

### B.8.1   GPT-4

1. **Problems released before GPT cutoff with functional correctness ∈ [0,0.0]**

   (a) **Subgroup sample 1 of 2**

   **title: scissors and tape** | **id: 1193_C** | **difficulty: 0** | **func_correct:** 0.0

   **url: https://codeforces.com/problemset/problem/1193/C**

   ```python
   def polygon_area(points):
       half_sum = 0
       n = len(points)
       for i in range(n):
           j = (i + 1) % n
           half_sum += (points[i][0] * points[j][1] - points[i][1] * points[j][0])
       return abs(half_sum) / 2

   def main():
       s = list(map(int, input().split()))[1:]
       t = list(map(int, input().split()))[1:]
       s_points = [(s[2 * i], s[2 * i + 1]) for i in range(len(s) // 2)]
       t_points = [(t[2 * i], t[2 * i + 1]) for i in range(len(t) // 2)]

       s_area = polygon_area(s_points)
       t_area = polygon_area(t_points)
       t_area_to_s_area = t_area / s_area

       # Assuming s_area = t_area.
       if abs(s_area - t_area) < 1e-6:
           print('scissors 0', len(s_points))
           for i in range(0, len(s), 2):
               print("1", s[i], s[i + 1], "0 0")
           print("tape", len(s_points), " ".join(map(str, range(1, len(s_points) + 1))))
           print("1", "0 0", " ".join(map(str, t)))
       else:
           print("ERROR: The input polygons do not have the same area.")

   if __name__ == "__main__":
       main()
   ```

   (b) **Subgroup sample 2 of 2**

   **title: string inside out** | **id: 530_B** | **difficulty: 1600** | **func_correct:** 0.0

   **url: https://codeforces.com/problemset/problem/530/B**

   ```python
   def manipulate_string(s):
       n = len(s) // 2
       return s[n - 1::-1] + s[:n - 1:-1]
   ```

2. **Problems released before GPT cutoff with functional correctness ∈ (0.0, 0.137]**

   (a) **Subgroup sample 1 of 2**

**title: zebra tower | id: 159_E | difficulty: 1700 | func_correct: 0.125**

**url: https://codeforces.com/problemset/problem/159/E**

```python
def main():
    n = int(input())
    cubes = [tuple(map(int, input().split())) for _ in range(n)]

    color_cubes = {}
    for i, (color, size) in enumerate(cubes):
        if color not in color_cubes:
            color_cubes[color] = []
        color_cubes[color].append((size, i))

    for color in color_cubes:
        color_cubes[color].sort(reverse=True)

    color_sums = [(sum(x[0] for x in color_cubes[c]), c) for c in color_cubes]
    color_sums.sort(reverse=True)

    max_height = 0
    best_tower = []
    for i in range(len(color_sums)):
        for j in range(i+1, len(color_sums)):
            height1, color1 = color_sums[i]
            height2, color2 = color_sums[j]
            tower = []

            while color_cubes[color1] and color_cubes[color2]:
                size1, idx1 = color_cubes[color1].pop()
                size2, idx2 = color_cubes[color2].pop()
                tower.append(idx1)
                tower.append(idx2)

            # If there is an extra cube of the first color, add it to the tower
            if color_cubes[color1]:
                size1, idx1 = color_cubes[color1].pop()
                tower.append(idx1)

            curr_height = sum(cubes[idx][1] for idx in tower)

            if curr_height > max_height:
                max_height = curr_height
                best_tower = tower

    print(max_height)
    print(len(best_tower))
    print(*[x+1 for x in best_tower])

if __name__ == "__main__":
    main()
```

(b) **Subgroup sample 2 of 2**

title: **dirty arkady's kitchen** | id: **827_F** | difficulty: **3200** | func_correct: 0.053

```python
def main():
    n, m = map(int, input().split())
    passages = [tuple(map(int, input().split())) for _ in range(m)]

    def can_reach(destination, time):
        visited = [False] * (n + 1)
        stack = [(1, time)]

        while stack:
            place, t = stack.pop()
            if place == destination:
                return True
            if visited[place]:
                continue
            visited[place] = True

            for a, b, l, r in passages:
                if t >= l and t < r:
                    next_place = a if b == place else (b if a == place else None)
                    if next_place is not None and not visited[next_place]:
                        stack.append((next_place, t + 1))

        return False

    start_time = 0
    while not can_reach(n, start_time):
        start_time += 1
        if start_time > 10**9:
            print(-1)
            return

    print(start_time)

main()
```

3. **Problems released before GPT cutoff with functional correctness $\in$ (0.137, 0.667]**

   (a) **Subgroup sample 1 of 2**

```
title: alarm clock | id: 898_D | difficulty: 1600 | func_correct: 0.316

url: https://codeforces.com/problemset/problem/898/D

def min_alarms_to_turn_off(n, m, k, alarms):
    alarms.sort()
    diffs = [alarms[i+1] − alarms[i] for i in range(len(alarms)−1)]
    diffs.sort()
    turn_off = 0
    while len(diffs) >= k−1 and sum(diffs[:k−1]) + k−1 < m:
        turn_off += 1
        diffs.pop()
    return turn_off

def main():
    n, m, k = map(int, input().split())
    alarms = list(map(int, input().split()))
    print(min_alarms_to_turn_off(n, m, k, alarms))

main()
```

(b) **Subgroup sample 2 of 2**

```
title: row of models | id: 887_F | difficulty: 2500 | func_correct: 0.424

url: https://codeforces.com/problemset/problem/887/F

def main():
    n, k = map(int, input().split())
    heights = list(map(int, input().split()))

    def test_swap(heights, k):
        first_higher = −1
        for i in range(len(heights) − 1):
            if heights[i] > heights[i + 1]:
                if first_higher == −1:
                    first_higher = i
                elif i − first_higher < k − 1:
                    return False
        return first_higher == −1 or first_higher >= k − 1

    if test_swap(heights, k) or test_swap(heights[::−1], k):
        print("YES")
    else:
        print("NO")

if __name__ == '__main__':
    main()
```

4. **Problems released before GPT cutoff with functional correctness ∈ (0.667, 1.0]**

   (a) **Subgroup sample 1 of 2**

title: **alternating current** | id: **343_B** | difficulty: **1600** | func_correct: 1.0

url: **https://codeforces.com/problemset/problem/343/B**

```python
def can_be_untangled(s):
    stack = []
    for c in s:
        if stack and stack[-1] == c:
            stack.pop()
        else:
            stack.append(c)
    return "Yes" if not stack else "No"

input_sequence = input().strip()
print(can_be_untangled(input_sequence))
```

(b) **Subgroup sample 2 of 2**

title: **date calculation** | id: **130_J** | difficulty: **1800** | func_correct: 1.0

url: **https://codeforces.com/problemset/problem/130/J**

```python
def is_leap_year(year):
    return year % 400 == 0 or (year % 4 == 0 and year % 100 != 0)

def find_date(year, day_num):
    days_in_months = [31, 28, 31, 30, 31, 30, 31, 31, 30, 31, 30, 31]
    if is_leap_year(year):
        days_in_months[1] = 29
    month = 0
    while day_num > days_in_months[month]:
        day_num -= days_in_months[month]
        month += 1
    return day_num, month + 1

year = int(input())
day_num = int(input())
day, month = find_date(year, day_num)
print(day, month)
```

5. **Problems released after GPT cutoff with functional correctness ∈ [0,0.0]**

(a) **Subgroup sample 1 of 2**

```
title: changing brackets | id: 1593_G | difficulty: 2200 | func_correct: 0.0
url: https://codeforces.com/problemset/problem/1593/G

def min_cost(s, l, r):
    cnt_open = cnt_bracket = 0
    cost = 0
    for i in range(l, r+1):
        if s[i] == '(' or s[i] == '[':
            cnt_open += 1
        else:
            cnt_open -= 1
            if cnt_open < 0:
                if s[i] == ')':
                    cnt_bracket += 1
                cnt_open = 0
    return cnt_bracket

t = int(input())
for _ in range(t):
    s = input()
    q = int(input())
    for _ in range(q):
        l, r = map(int, input().split())
        print(min_cost(s, l-1, r-1))
```

(b) **Subgroup sample 2 of 2**

```
title: premutation | id: 1790_C | difficulty: 1000 | func_correct: 0.0
url: https://codeforces.com/problemset/problem/1790/C

def reconstruct_permutation(t, test_cases):
    def restore_permutation(n, sequences):
        occurrences = [0] * (n + 1)
        for seq in sequences:
            for num in seq:
                occurrences[num] += 1
        return [i for i, count in enumerate(occurrences) if count == n - 1]

    results = []
    for case in test_cases:
        n, sequences = case
        results.append(restore_permutation(n, sequences))
    return results

t = 5
test_cases = [
    (4, [[4, 2, 1], [4, 2, 3], [2, 1, 3], [4, 1, 3]]),
    (3, [[3, 2], [3, 1], [1, 2]]),
    (5, [[4, 2, 1, 3], [2, 1, 3, 5], [4, 2, 3, 5], [4, 1, 3, 5], [4, 2, 1, 5]]),
    (3, [[1, 3], [2, 3], [1, 2]]),
    (4, [[3, 1, 2], [1, 3, 4], [2, 1, 3], [1, 2, 4]])
]

result = reconstruct_permutation(t, test_cases)
for r in result:
    print(*r)
```

6. **Problems released after GPT cutoff with functional correctness ∈ (0.0, 0.137]**

   (a) **Subgroup sample 1 of 2**

title: arranging crystal balls | id: 1740_I | difficulty: 3500 | func_correct: 0.062

url: https://codeforces.com/problemset/problem/1740/I

```python
def gcd(a, b):
    while b:
        a, b = b, a % b
    return a

def minimum_operations(n, m, k, a):
    gcd_ak = a[0] - a[k]
    for i in range(k, n):
        gcd_ak = gcd(gcd_ak, a[i] - a[i - k])
    if gcd_ak % m != 0:
        return -1
    return sum(abs(a[i] - a[0]) // m for i in range(n)) // k

n, m, k = map(int, input().split())
a = list(map(int, input().split()))
print(minimum_operations(n, m, k, a))
```

(b) **Subgroup sample 2 of 2**

```python
def main():
    MOD = 998244353
    n = int(input())
    desired_votes = list(map(int, input().split()))
    team = list(map(int, input().split()))

    team_votes = [0] * (n + 1)
    for tv in team:
        team_votes[tv] += 1

    dp = [[0] * (n + 1) for _ in range(n + 1)]
    dp[0][0] = 1
    for i in range(1, n + 1):
        for j in range(n + 1):
            dp[i][j] = dp[i - 1][j]
            if j - desired_votes[i - 1] >= 0:
                dp[i][j] += dp[i - 1][j - desired_votes[i - 1]]
                if dp[i][j] >= MOD:
                    dp[i][j] -= MOD

    ans = 1
    total_non_team_votes = 0
    for i in range(1, n + 1):
        count = team_votes[i]
        if count == 0:
            continue
        tmp = dp[total_non_team_votes + count][count] * ans
        if tmp >= MOD:
            tmp %= MOD
        ans = tmp
        total_non_team_votes += count

    print(ans)

main()
```

7. **Problems released after GPT cutoff with functional correctness $\in$ (0.137, 0.667]**

   (a) **Subgroup sample 1 of 2**

title: count binary strings | id: 1767_C | difficulty: 2100 | func_correct: 0.364

url: https://codeforces.com/problemset/problem/1767/C

```python
def main():
    MOD = 998244353

    n = int(input())
    a = [list(map(int, input().split())) for _ in range(n)]

    dp = [[[0] * (n + 1) for _ in range(n + 1)] for _ in range(3)]

    for i in range(n+1):
        dp[0][i][i] = dp[1][i][i] = 1

    for l in range(2, n+1):
        for i in range(1, n - l + 2):
            j = i + l - 1

            if a[i - 1][j - i] != 2:
                dp[0][i][j] += (dp[0][i + 1][j] + dp[1][i + 1][j]) % MOD
                dp[0][i][j] %= MOD
            if a[i - 1][j - i] != 1:
                dp[1][i][j] += (dp[0][i + 1][j] + dp[1][i + 1][j]) % MOD
                dp[1][i][j] %= MOD

            for k in range(i, j):
                if a[i - 1][k - i] != 1 and a[k + 1 - 1][j - (k + 1)] != 1:
                    dp[2][i][j] += dp[2][i][k] * dp[2][k + 1][j] % MOD
                if a[i - 1][k - i] != 1 and a[k + 1 - 1][j - (k + 1)] != 2:
                    dp[2][i][j] += dp[2][i][k] * dp[1][k + 1][j] % MOD
                if a[i - 1][k - i] != 2 and a[k + 1 - 1][j - (k + 1)] != 1:
                    dp[2][i][j] += dp[0][i][k] * dp[2][k + 1][j] % MOD
                if a[i - 1][k - i] != 2 and a[k + 1 - 1][j - (k + 1)] != 2:
                    dp[2][i][j] += dp[0][i][k] * dp[1][k + 1][j] % MOD

            dp[2][i][j] %= MOD

    print(dp[2][1][n])

if __name__ == "__main__":
    main()
```

(b) **Subgroup sample 2 of 2**

```
title: interesting sum | id: 1720_B | difficulty: 800 | func_correct: 0.5
url: https://codeforces.com/problemset/problem/1720/B

def maximum_beauty(t, test_cases):
    results = []
    for n, a in test_cases:
        max_outside = max(a)
        max_inside, min_inside = max(a[1:-1]), min(a[1:-1])
        ans = max(max_outside - a[0] + max_inside - a[-1],
                  max_outside - a[-1] + max_inside - a[0])
        results.append(ans)
    return results

def main():
    t = int(input().strip())
    test_cases = []
    for _ in range(t):
        n = int(input().strip())
        a = list(map(int, input().strip().split()))
        test_cases.append((n, a))
    output = maximum_beauty(t, test_cases)
    for res in output:
        print(res)

if __name__ == '__main__':
    main()
```

8. **Problems released after GPT cutoff with functional correctness ∈ (0.667, 1.0]**

    (a) **Subgroup sample 1 of 2**

```
title: the boss can count pairs | id: 1830_B | difficulty: 2000 | func_correct:
1.0
url: https://codeforces.com/problemset/problem/1830/B

def main():
    t = int(input().strip())
    for _ in range(t):
        n = int(input().strip())
        a = list(map(int, input().strip().split()))
        b = list(map(int, input().strip().split()))

        count = 0
        for i in range(n):
            for j in range(i + 1, n):
                if a[i] * a[j] == b[i] + b[j]:
                    count += 1
        print(count)

if __name__ == "__main__":
    main()
```

    (b) **Subgroup sample 2 of 2**

```
title: interesting sections | id: 1609_F | difficulty: 2800 | func_correct: 1.0
url: https://codeforces.com/problemset/problem/1609/F

def main():
    n = int(input().strip())
    a = list(map(int, input().strip().split()))

    def bit_count(x):
        return bin(x).count('1')

    ans = 0
    for l in range(n):
        min_val = a[l]
        max_val = a[l]
        for r in range(l, n):
            min_val = min(min_val, a[r])
            max_val = max(max_val, a[r])
            if bit_count(min_val) == bit_count(max_val):
                ans += 1
    print(ans)

if __name__ == '__main__':
    main()
```

1. **Problems released before GPT cutoff with functional correctness ∈ [0,0.0]**

    (a) **Subgroup sample 1 of 2**

```python
n, q = map(int, input().split())
a = sorted(map(int, input().split()))
p = {a[i]:i for i in range(n)}
s = [x−y for x,y in zip(a[1:], a)]
s1 = [x−y for x,y in zip(a[2:], a)]
ans1 = ans2 = sum(s)
for _ in range(q):
    t, x = map(int, input().split())
    x −= 1
    if t == 1:
        a.append(x)
        a.sort()
        i = p[x] = a.index(x)
        if i > 0: ans1 += max(0, a[i]−a[i−1]−1)
        if i < n−1: ans1 += max(0, a[i+1]−a[i]−1)
        if i > 1: ans1 −= max(0, a[i]−a[i−2]−1) − s1[i−2]
        if i < n−2: ans1 −= max(0, a[i+2]−a[i+1]−1) − s[i−1]
        n += 1
        s1, s = s, [x−y for x,y in zip(a[1:], a)]
    else:
        i = p[x]
        if i > 0: ans1 −= max(0, a[i]−a[i−1]−1)
        if i < n−1: ans1 −= max(0, a[i+1]−a[i]−1)
        if i > 1: ans1 += max(0, a[i]−a[i−2]−1) − s1[i−2]
        if i < n−2: ans1 += max(0, a[i+2]−a[i+1]−1) − s[i−1]
        a.remove(x)
        n −= 1
        del p[x]
        s1, s = s, [x−y for x,y in zip(a[1:], a)]
    print(ans1)
print(ans2)
```

(b) **Subgroup sample 2 of 2**

**title: boboniu and string** | **id: 1394_C** | **difficulty: 2600** | **func_correct:** 0.0

**url: https://codeforces.com/problemset/problem/1394/C**

```python
from collections import Counter

def can_reach(target, source):
    """Check if target can be obtained from source by removing 'B','N' and 'BN'/'
        ↪ NB'."""
    n = len(target)
    m = len(source)
    i = j = 0
    while i < n and j < m:
        if target[i] == source[j]:
            i += 1
            j += 1
        elif target[i] == 'B' and source[j] == 'N':
            j += 1
        elif target[i] == 'N' and source[j] == 'B':
            j += 1
        elif target[i:i+2] == 'BN' and (j+1 < m and source[j:j+2] == 'BN'):
            j += 2
            i += 2
        elif target[i:i+2] == 'NB' and (j+1 < m and source[j:j+2] == 'NB'):
            j += 2
            i += 2
        else:
            return False
    return i == n

n = int(input())
strings = []
for i in range(n):
    strings.append(input().strip())

freq = Counter(''.join(strings))
t = ''
if freq['B'] >= freq['N']:
    t += 'B' * freq['B']
else:
    t += 'N' * freq['N']

def binary_search():
    """Use binary search to find the minimum value of max_dist."""
    l, r = 1, 2 * 10 ** 6
    while l < r:
        mid = (l + r) // 2
        can_make = False
        for char in ['B', 'N', 'BN', 'NB']:
            for start in range(len(t)):
                new_t = t[:start] + char + t[start:]
                if all(can_reach(new_t, s) for s in strings):
                    can_make = True
                    break
            if can_make:
                break
        if can_make:
            r = mid
        else:
            l = mid + 1
    return l
# Note from authors: cutting this submission early for display convenience (it's too
    ↪ long to show all of it).
```

2. **Problems released before GPT cutoff with functional correctness ∈ (0.0, 0.051]**

   (a) **Subgroup sample 1 of 2**

   > **title: up the strip** | **id: 1558_B** | **difficulty: 1900** | **func_correct:** 0.03
   >
   > **url: https://codeforces.com/problemset/problem/1558/B**
   >
   > ```python
   > n, m = map(int, input().split())
   > dp = [0] * (n+1)
   > dp[n] = 1
   > for i in range(n−1, 0, −1):
   >     dp[i] = (dp[i+1] + dp[i∗2]) % m
   > print(dp[1])
   > ```

   (b) **Subgroup sample 2 of 2**

   > **title: yellow cards** | **id: 1215_A** | **difficulty: 1000** | **func_correct:** 0.023
   >
   > **url: https://codeforces.com/problemset/problem/1215/A**
   >
   > ```python
   > a1 = int(input())
   > a2 = int(input())
   > k1 = int(input())
   > k2 = int(input())
   > n = int(input())
   >
   > min_players_out = max(0, n − (a1∗(k1−1) + a2∗(k2−1)))
   > max_players_out = min(n, min(a1,k1)∗a1 + min(a2,k2)∗a2 − n)
   >
   > print(min_players_out, max_players_out)
   > ```

3. **Problems released before GPT cutoff with functional correctness ∈ (0.051, 0.4]**

   (a) **Subgroup sample 1 of 2**

   > **title: exchange of books** | **id: 683_H** | **difficulty: 1900** | **func_correct:** 0.1
   >
   > **url: https://codeforces.com/problemset/problem/683/H**
   >
   > ```python
   > n, k = map(int, input().split())
   > friends = list(map(int, input().split()))
   > start = list(range(1, n+1)) # starting arrangement of books
   >
   > # create a dict to map current position to the next position after one day
   > next_pos = {i+1: friend for i, friend in enumerate(friends)}
   >
   > # simulate book exchange for k days
   > for i in range(k):
   >     end = [0] * n # initialize empty list to hold end arrangement of books
   >     for j in range(n):
   >         end[next_pos[j+1]−1] = start[j] # exchange books
   >     start = end # update starting arrangement to end arrangement
   >     next_pos = {j+1: next_pos[next_pos[j+1]] for j in range(n)} # update dict
   >
   > # print final arrangement of books
   > print(∗start)
   > ```

   (b) **Subgroup sample 2 of 2**

```
n, m, k = map(int, input().split())
alarms = sorted(map(int, input().split()))
interval = [0] * (10**6 + 1)
for i in range(1, n − k + 2):
    start = alarms[i−1]
    end = alarms[i+k−2] if i+k−1 <= n else 10**6
    interval[start:end+1] = [j+1 for j in interval[start:end+1]]
print(sum([1 for x in interval if x >= m]))
```

4. **Problems released before GPT cutoff with functional correctness ∈ (0.4, 1.0]**

   (a) **Subgroup sample 1 of 2**

**title: chemistry in berland** | **id: 846_E** | **difficulty: 2300** | **func_correct:** 0.875

```
n = int(input())
b = list(map(int, input().split()))
a = list(map(int, input().split()))
adj = [[] for _ in range(n)]
for i in range(1, n):
    x, k = map(int, input().split())
    adj[i].append((x−1, k))
    adj[x−1].append((i, k))
visited = [False] * n
def dfs(u):
    visited[u] = True
    for v, k in adj[u]:
        if not visited[v]:
            a[v] −= k * a[u]
            b[v] −= k * b[u]
            if a[v] > b[v] or a[v] % gcd(a[u], b[u]) != 0:
                return False
            if not dfs(v):
                return False
    return True
def gcd(a, b):
    while b:
        a, b = b, a % b
    return a
for i in range(n):
    if not visited[i] and a[i] != 0:
        if dfs(i):
            continue
        else:
            print("NO")
            break
else:
    print("YES")
```

   (b) **Subgroup sample 2 of 2**

title: **optimal subsequences (easy version)** | id: **1227_D1** | difficulty: **1600** | func_correct: 0.5

```python
n = int(input())
a = list(map(int, input().split()))
m = int(input())

# Indices of maximum elements in all prefixes of a
max_indices = [0] * n
cur_max_idx = 0
for i in range(n):
    if a[i] > a[cur_max_idx]:
        cur_max_idx = i
    max_indices[i] = cur_max_idx

for _ in range(m):
    k, pos = map(int, input().split())
    subseq = []
    i = max_indices[k − 1] # start with the maximum element in the k−length prefix
    while len(subseq) < k:
        if a[i] != 0:
            subseq.append(a[i])
        i += 1
        if i >= n: # end of a is reached
            break
    subseq.sort(reverse=True) # lexicographically minimal subsequence with
        ↪ maximum sum
    print(subseq[pos − 1])
```

5. **Problems released after GPT cutoff with functional correctness ∈ [0,0.0]**

   (a) **Subgroup sample 1 of 2**

**title: anti-fibonacci permutation** | **id: 1644_B** | **difficulty: 800** |
**func_correct:** 0.0

**url: https://codeforces.com/problemset/problem/1644/B**

```python
# function to generate a permutation
def generate_permutation(n):
    return list(range(1,n+1))

# function to check if a permutation is anti−fibonacci
def is_anti_fibonacci(p):
    for i in range(2,len(p)):
        if p[i−2]+p[i−1] == p[i]:
            continue
        else:
            return False
    return True

t = int(input())
for _ in range(t):
    n = int(input())
    # create a list of all possible permutations
    all_permutations = [generate_permutation(n)]
    # iterate over all permutations and check if they are anti−fibonacci
    for i in range(2, n):
        for j in range(i):
            new_permutations = []
            for p in all_permutations:
                new_p = p.copy()
                new_p[i], new_p[j] = new_p[j], new_p[i]
                if is_anti_fibonacci(new_p):
                    new_permutations.append(new_p)
            all_permutations += new_permutations
    # print the first n anti−fibonacci permutations
    for p in all_permutations[:n]:
        print(∗p)
```

(b) **Subgroup sample 2 of 2**

**title: cypher** | **id: 1703_C** | **difficulty: 800** | **func_correct:** 0.0

**url: https://codeforces.com/problemset/problem/1703/C**

```python
def original_sequence(n, final, moves):
    # initialize original sequence to all zeroes
    orig = [0]*n

    # apply moves in reverse order to determine original sequence
    for i in range(n-1, -1, -1):
        if moves[i][0] == 'D':
            shift = int(moves[i][1:])
        else:
            shift = -int(moves[i][1:])

        # adjust for wheel rollover
        if final[i] + shift < 0:
            shift += 10
        elif final[i] + shift > 9:
            shift -= 10

        orig[i] = (final[i] + shift) % 10

    return orig

t = int(input())
for _ in range(t):
    n = int(input())
    final = [int(x) for x in input().split()]
    moves = [input().strip() for _ in range(n)]

    orig = original_sequence(n, final, moves)
    print(*orig)
```

6. **Problems released after GPT cutoff with functional correctness ∈ (0.0, 0.051]**

   (a) **Subgroup sample 1 of 2**

   **title: torus path** | **id: 1765_K** | **difficulty: 1500** | **func_correct:** 0.04

   **url: https://codeforces.com/problemset/problem/1765/K**

```python
n = int(input())
grid = []
for i in range(n):
    grid.append(list(map(int, input().split())))
dp = [[0]*n for i in range(n)]
dp[0][0] = grid[0][0]
for i in range(1,n):
    dp[i][0] = dp[i-1][0] + grid[i][0]
for j in range(1,n):
    dp[0][j] = dp[0][j-1] + grid[0][j]
for i in range(1,n):
    for j in range(1,n):
        dp[i][j] = max(dp[i-1][j], dp[i][j-1]) + grid[i][j]
print(dp[-1][-1])
```

   (b) **Subgroup sample 2 of 2**

title: **olympic team building** | id: **1779_H** | difficulty: **3500** | func_correct: **0.023**

url: **https://codeforces.com/problemset/problem/1779/H**

```python
n = int(input())
strengths = list(map(int, input().split()))

winners = [True] * n # Assume all players are initially winners

while n > 1:
    new_winners = [False] * n
    for i in range(0, n, 2):
        total_strength = strengths[i] + strengths[i+1]
        if total_strength == max(total_strength, 0): # If both strengths are positive or
            #  only one is negative
            new_winners[i] = True
        else:
            new_winners[i+1] = True
    winners = [winners[i] and new_winners[i] for i in range(n)] # Update winners
        #  list
    n //= 2 # Half the number of players
    strengths = [strengths[2*i] + strengths[2*i+1] for i in range(n)] # Sum strengths
        #  for remaining players

for i in range(len(winners)):
    if winners[i]:
        print("1", end="")
    else:
        print("0", end="")
```

7. **Problems released after GPT cutoff with functional correctness ∈ (0.051, 0.4]**

   (a) **Subgroup sample 1 of 2**

**title:** magician and pigs (hard version) | **id:** 1774_F2 | **difficulty:** 2700 |
**func_correct:** 0.375

**url:** https://codeforces.com/problemset/problem/1774/F2

```python
n = int(input())

# create an empty list to store the pigs' health points
pigs = []

# iterate through n operations
for i in range(n):
    op = input().split()
    if op[0] == '1':
        # create a new pig with x health points
        pigs.append(int(op[1]))
    elif op[0] == '2':
        # reduce all pigs' health points by x
        for j in range(len(pigs)):
            pigs[j] -= int(op[1])
    else:
        # repeat all previous operations (including repeats)
        for j in range(i):
            prev_op = input().split()
            if prev_op[0] == '1':
                # create a new pig with x health points
                pigs.append(int(prev_op[1]))
            elif prev_op[0] == '2':
                # reduce all pigs' health points by x
                for k in range(len(pigs)):
                    pigs[k] -= int(prev_op[1])

    # remove dead pigs from the list
    pigs = [p for p in pigs if p > 0]

# print the number of living pigs modulo 998244353
print(len(pigs) % 998244353)
```

(b) **Subgroup sample 2 of 2**

**title:** physical examination | **id:** 1571_I | **difficulty:** 3200 | **func_correct:** 0.333

**url:** https://codeforces.com/problemset/problem/1571/I

```python
for _ in range(int(input())):
    n = int(input())
    L = list(map(int, input().split()))
    R = list(map(int, input().split()))
    x = max(0, max(L) - min(R))
    p = [0] * n
    for i in range(n):
        p[i] = i
    p.sort(key=lambda i: L[i])
    current_time = x
    for i in range(n):
        if L[p[i]] > current_time + i or R[p[i]] < current_time + i:
            print(-1)
            break
        print(current_time + i, end=' ')
    else:
        print()
        print(*[p[i]+1 for i in range(n)])
```

8. **Problems released after GPT cutoff with functional correctness $\in$ (0.4, 1.0]**

    (a) **Subgroup sample 1 of 2**

**title:** replace the numbers | **id:** 1620_E | **difficulty:** 1900 | **func_correct:** 1.0

**url:** https://codeforces.com/problemset/problem/1620/E

```python
q = int(input())
arr = []
for i in range(q):
    query = input().split()
    if query[0] == '1':
        arr.append(int(query[1]))
    else:
        x = int(query[1])
        y = int(query[2])
        for j in range(len(arr)):
            if arr[j] == x:
                arr[j] = y
print(*arr)
```

    (b) **Subgroup sample 2 of 2**

```python
def min_operations(s):
    colors = set(s)
    if len(colors) < 4:
        return -1
    on = set()
    count = 0
    for i in range(4):
        if s[i] not in on:
            on.add(s[i])
            count += 1
    return count

t = int(input())
for i in range(t):
    s = input()
    print(min_operations(s))
```

Table 1: Regression table for Pass Rate of GPT-4 on the Codeforces dataset. See Figure 9 for plot of regression coefficients.

| | Dependent variable: | |
| --- | --- | --- |
| | Pass Rate | |
| | Before Cutoff | After Cutoff |
| | (1) | (2) |
| Difficulty | 0.084 | 0.204 |
| | (0.082, 0.087) | (0.183, 0.228) |
| | p = 0.000* | p = 0.000* |
| | | |
| GitHub_Presence | 1.044 | 1.000 |
| | (1.038, 1.050) | (0.986, 1.014) |
| | p = 0.000* | p = 0.988 |
| | | |
| Constant | 37,810,359.000 | 36,517.320 |
| | (29,642,418.000, 48,261,514.000) | (15,107.710, 88,780.010) |
| | p = 0.000* | p = 0.000* |
| | | |
| Observations | 6,693 | 1,378 |
| Log Likelihood | −61,571.750 | −3,358.241 |
| Akaike Inf. Crit. | 123,149.500 | 6,722.482 |
| Note: | | *p<0.05 |

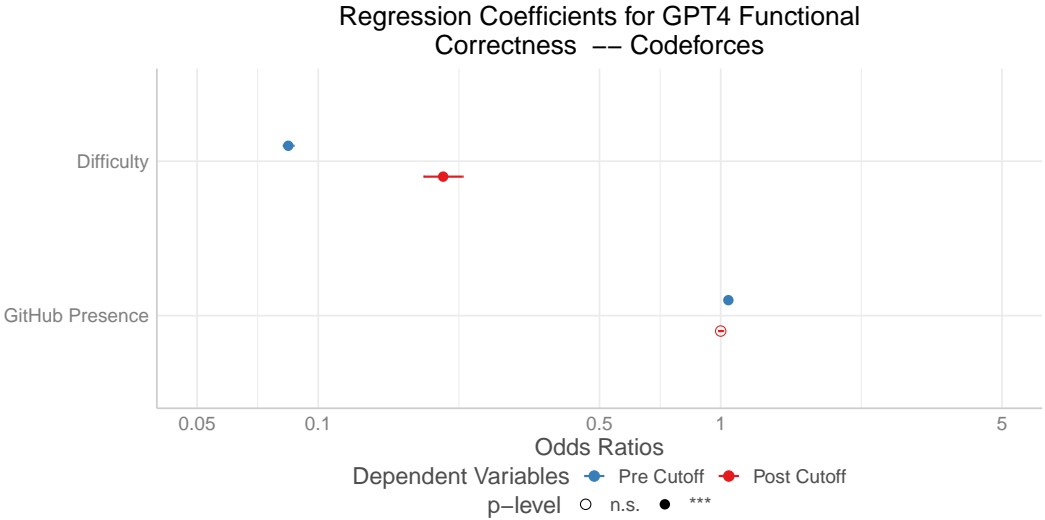

Figure 9: Regression coefficients plots of Pass Rate for GPT-4 on the Project Codeforces Dataset. See Table 1 for regression coefficients.

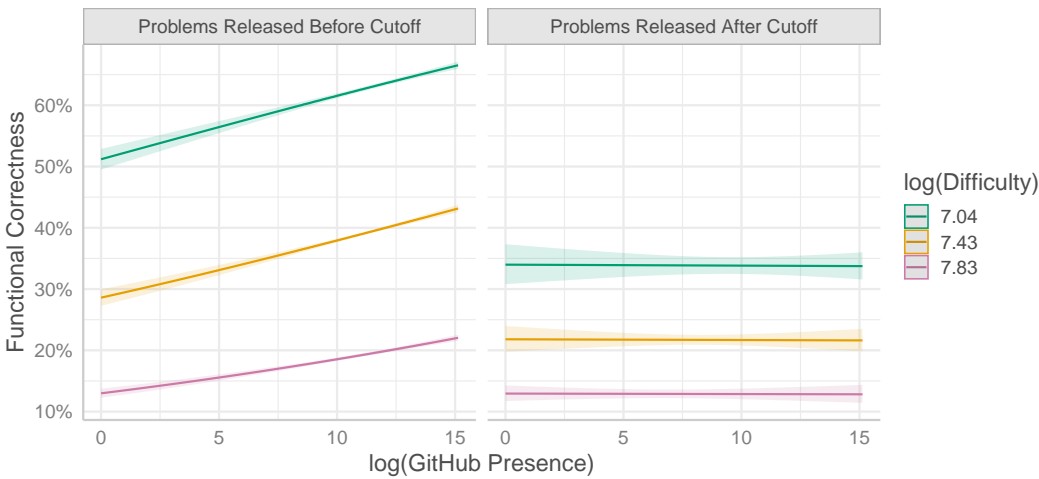

Figure 10: Marginal Effects of Pass Rate for GPT-4 on the Codeforces Dataset

Table 2: Regression table for Pass Rate of GPT-3.5-Turbo on the Codeforces dataset. See Figure 11 for plot of regression coefficients.

|  | *Dependent variable:* | |
|---|---|---|
|  | Pass Rate | |
|  | Before Cutoff | After Cutoff |
|  | (1) | (2) |
| Difficulty | 0.115 | 0.228 |
|  | (0.112, 0.119) | (0.202, 0.257) |
|  | p = 0.000* | p = 0.000* |
| GitHub_Presence | 1.023 | 1.014 |
|  | (1.017, 1.030) | (0.998, 1.030) |
|  | p = 0.000* | p = 0.081 |
| Constant | 2,859,990.000 | 9,989.318 |
|  | (2,244,055.000, 3,646,978.000) | (3,842.640, 26,109.620) |
|  | p = 0.000* | p = 0.000* |
| Observations | 6,692 | 1,378 |
| Log Likelihood | −52,913.820 | −2,611.597 |
| Akaike Inf. Crit. | 105,833.600 | 5,229.195 |
| *Note:* | | *p<0.05 |

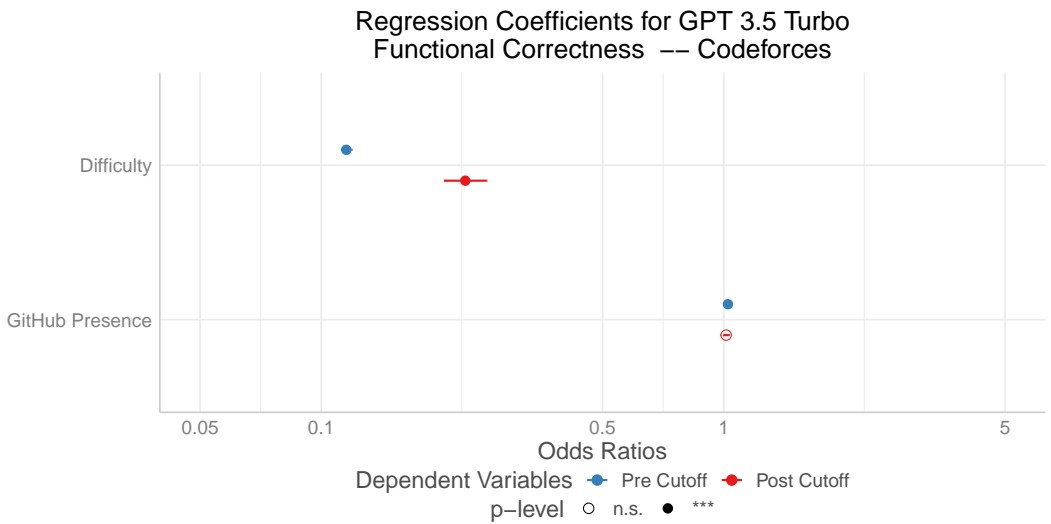

Figure 11: Regression coefficients plots of Pass Rate for GPT-3.5-Turbo on the Project Codeforces Dataset. See Table 2 for regression coefficients.

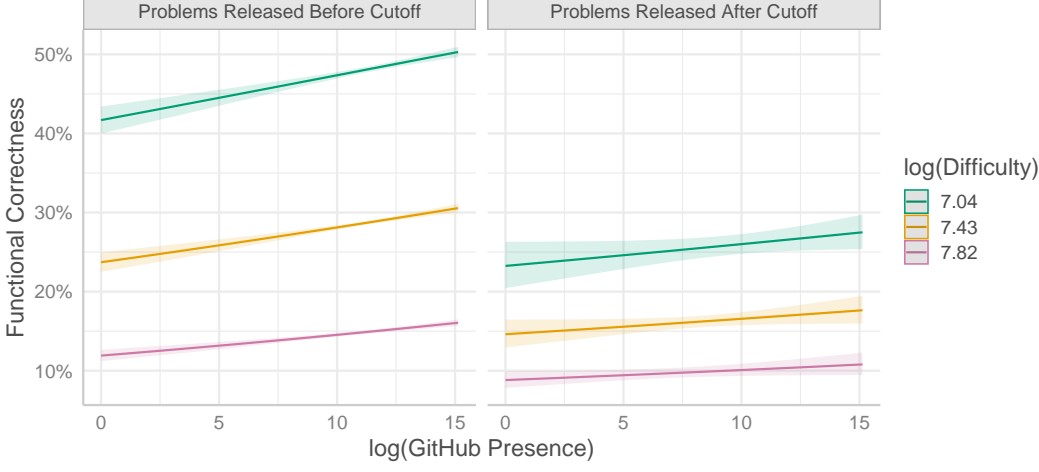

Figure 12: Marginal Effects of Pass Rate for GPT-3.5-Turbo on the Codeforces Dataset

Table 3: Code Bison Functional Correctness – Codeforces

| | *Dependent variable:* | |
|---|---|---|
| | Pass Rate | |
| | Before Cutoff | After Cutoff |
| | (1) | (2) |
| Difficulty | 0.221 | 0.344 |
| | (0.214, 0.228) | (0.240, 0.492) |
| | p = 0.000* | p = 0.000* |
| | | |
| Github_Presence | 1.033 | 1.023 |
| | (1.027, 1.040) | (0.978, 1.071) |
| | p = 0.000* | p = 0.316 |
| | | |
| Constant | 12,033.200 | 229.971 |
| | (9,387.552, 15,428.990) | (14.438, 3,694.907) |
| | p = 0.000* | p = 0.0002* |
| Observations | 7,807 | 217 |
| Log Likelihood | −38,944.730 | −228.904 |
| Akaike Inf. Crit. | 77,895.470 | 463.807 |
| *Note:* | *p<0.05; **p<[0.**]; ***p<[0.***] | |

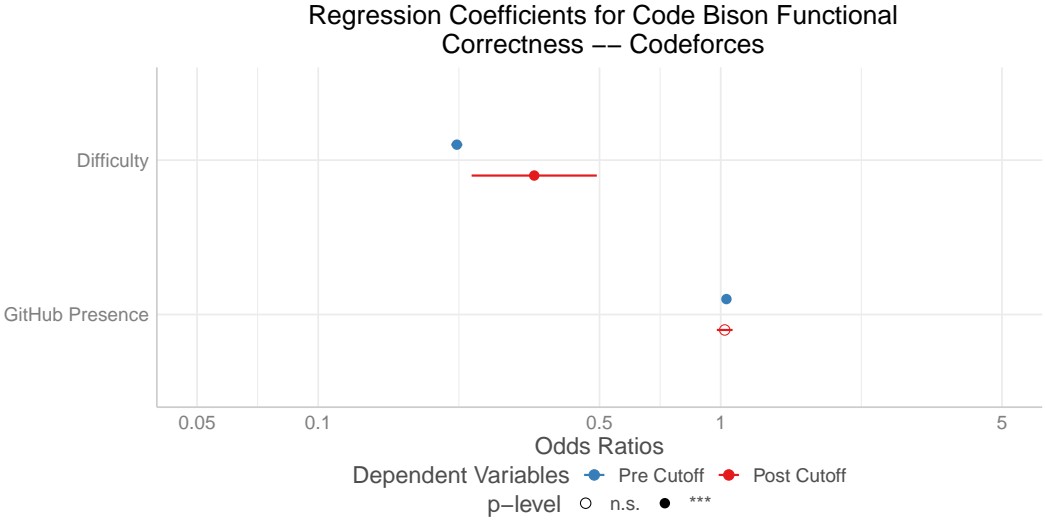

Figure 13: Regression coefficients plots of Pass Rate for Code Bison on the Project Codeforces Dataset. See Table 3 for regression coefficients.

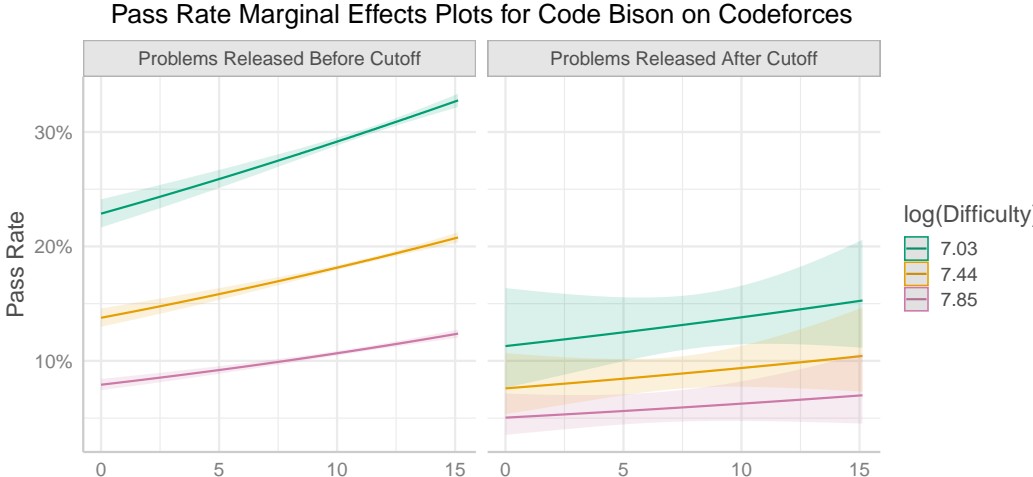

Figure 14: Marginal Effects of Pass Rate for Code Bison on the Codeforces Dataset

Table 4: Regression table for Pass Rate of GPT-4 on the Codeforces dataset (public test cases only). See Figure 15 for plot of regression coefficients.

|  | *Dependent variable:* | |
|---|---|---|
|  | Pass Rate (Public) | |
|  | Before Cutoff | After Cutoff |
|  | (1) | (2) |
| Difficulty | 0.122 | 0.834 |
|  | (0.110, 0.135) | (0.624, 1.116) |
|  | p = 0.000* | p = 0.221 |
|  |  |  |
| GitHub_Presence | 1.043 | 0.998 |
|  | (1.024, 1.063) | (0.964, 1.033) |
|  | p = 0.00001* | p = 0.895 |
|  |  |  |
| Constant | 2,468,839.000 | 0.647 |
|  | (1,118,457.000, 5,484,288.000) | (0.064, 6.445) |
|  | p = 0.000* | p = 0.712 |
|  |  |  |
| Observations | 6,693 | 1,378 |
| Log Likelihood | −7,352.622 | −712.065 |
| Akaike Inf. Crit. | 14,711.250 | 1,430.131 |
| *Note:* | | *p<0.05 |

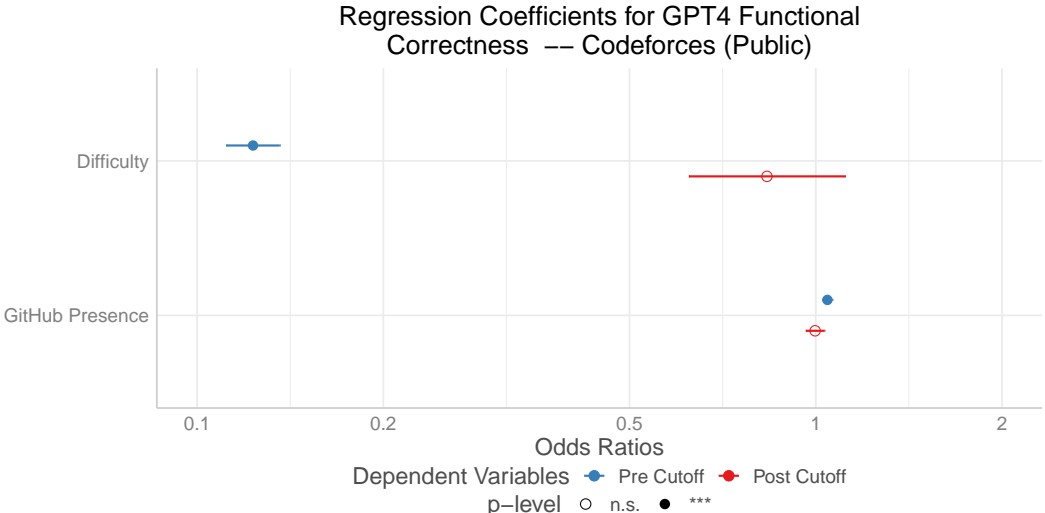

Figure 15: Regression coefficients plots of Pass Rate for GPT-4 on the Project Codeforces Dataset (evaluated on public test cases only). See Table 4 for regression coefficients.

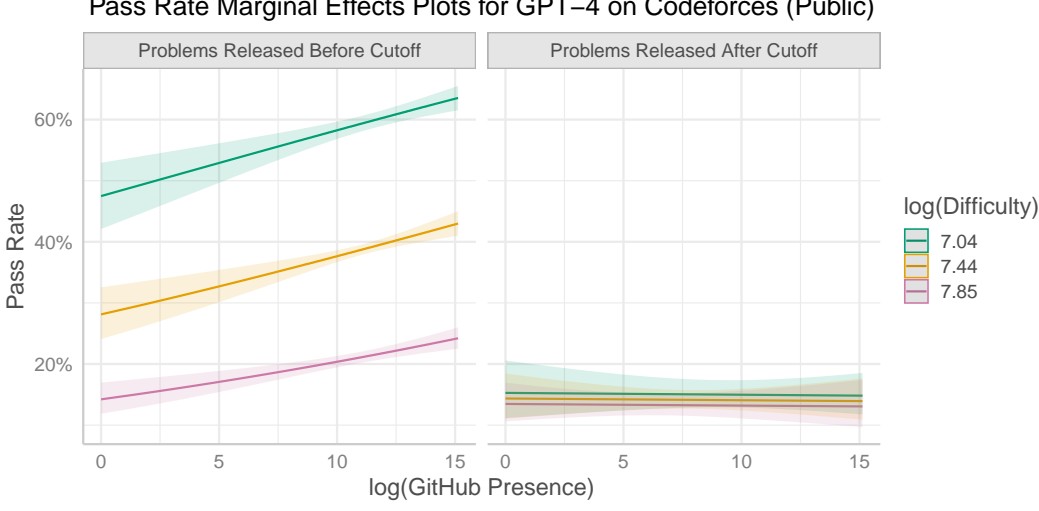

Figure 16: Marginal Effects of Pass Rate for GPT-4 on the Codeforces Dataset (evaluated on public test cases only)

Table 5: Regression table for Pass Rate of GPT-3.5-Turbo on the Codeforces dataset (public test cases only). See Figure 17 for plot of regression coefficients.

| | Dependent variable: | |
| --- | --- | --- |
| | Pass Rate (Public) | |
| | Before Cutoff | After Cutoff |
| | (1) | (2) |
| Difficulty | 0.157 | 1.046 |
| | (0.141, 0.173) | (0.744, 1.478) |
| | p = 0.000* | p = 0.798 |
| | | |
| GitHub_Presence | 1.027 | 0.998 |
| | (1.006, 1.047) | (0.960, 1.038) |
| | p = 0.011* | p = 0.919 |
| | | |
| Constant | 269,429.200 | 0.080 |
| | (119,822.600, 608,823.800) | (0.005, 1.190) |
| | p = 0.000* | p = 0.069 |
| | | |
| Observations | 6,693 | 1,378 |
| Log Likelihood | −6,461.524 | −562.326 |
| Akaike Inf. Crit. | 12,929.050 | 1,130.652 |
| *Note:* | | *p<0.05 |

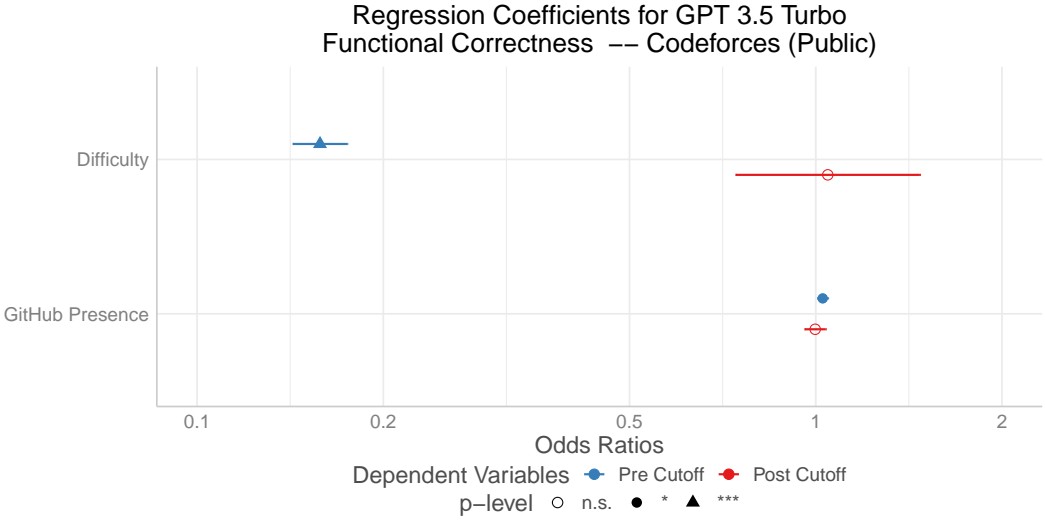

Figure 17: Regression coefficients plots of Pass Rate for GPT-3.5-Turbo on the Project Codeforces Dataset (evaluated on public test cases only). See Table 5 for regression coefficients.

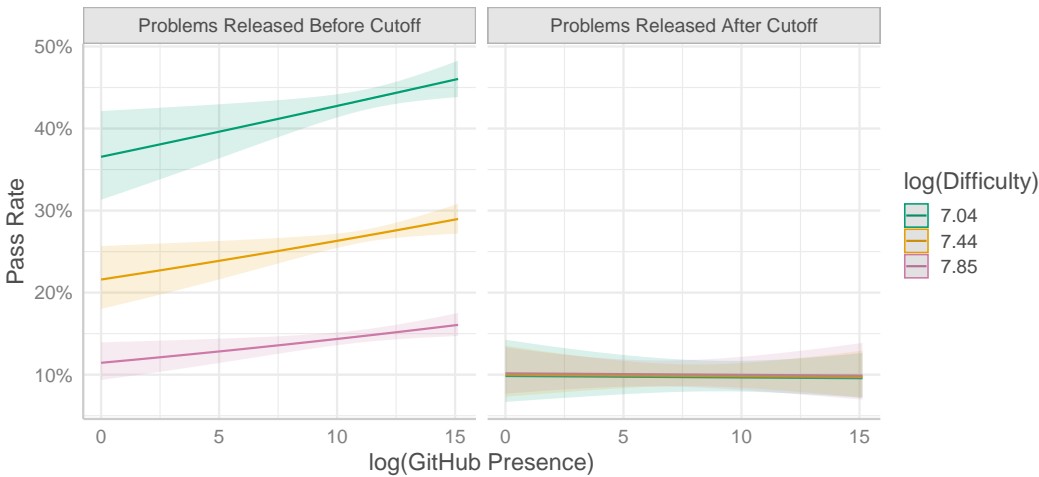

Figure 18: Marginal Effects of Pass Rate for GPT-3.5-Turbo on the Codeforces Dataset (evaluated on public test cases only)

Table 6: Code Bison Functional Correctness – Codeforces (Public)

|  | *Dependent variable:* | |
|---|---|---|
|  | Pass Rate | |
|  | Before Cutoff | After Cutoff |
|  | (1) | (2) |
| Difficulty | 0.309 | 0.606 |
|  | (0.278, 0.343) | (0.137, 2.627) |
|  | p = 0.000* | p = 0.498 |
| Github_Presence | 1.049 | 0.886 |
|  | (1.029, 1.069) | (0.741, 1.034) |
|  | p = 0.00001* | p = 0.147 |
| Constant | 742.217 | 3.558 |
|  | (322.484, 1,710.418) | (0.00003, 400,997.100) |
|  | p = 0.000* | p = 0.827 |
| Observations | 7,807 | 217 |
| Log Likelihood | −5,463.652 | −40.781 |
| Akaike Inf. Crit. | 10,933.300 | 87.561 |
| *Note:* | *p<0.05; **p<[0.**]; ***p<[0.***] | |

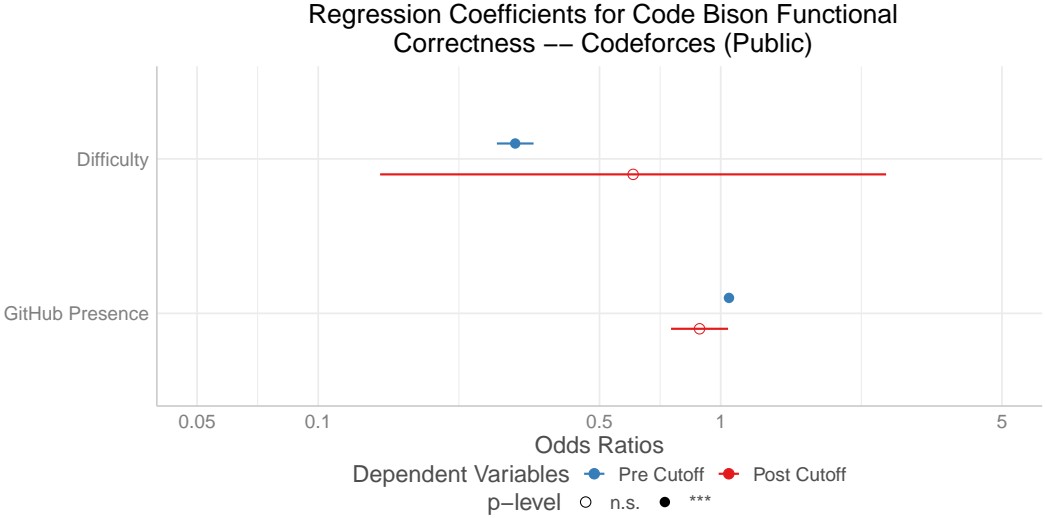

Figure 19: Regression coefficients plots of Pass Rate for Code Bison on the Project Codeforces Dataset (evaluated on public test cases only). See Table 6 for regression coefficients.

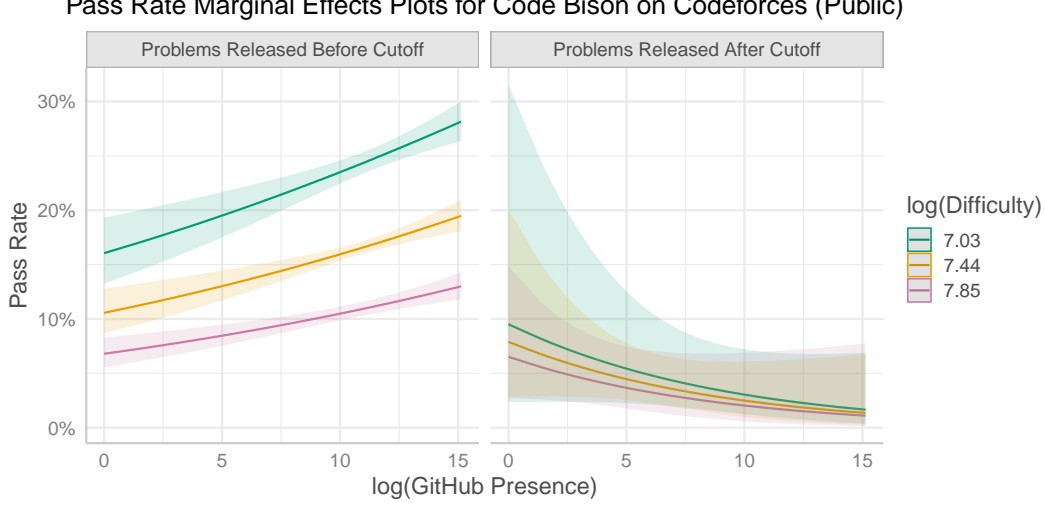

Figure 20: Marginal Effects of Pass Rate for Code Bison on the Codeforces Dataset (evaluated on public test cases only)

Table 7: Regression table for Pass Rate of GPT-4 on the Codeforces dataset (private test cases only). See Figure 21 for plot of regression coefficients.

|  | Dependent variable: | |
| --- | --- | --- |
|  | Pass Rate (Private) | |
|  | Before Cutoff | After Cutoff |
|  | (1) | (2) |
| Difficulty | 0.081 | 0.147 |
|  | (0.079, 0.084) | (0.129, 0.166) |
|  | $p = 0.000^*$ | $p = 0.000^*$ |
| GitHub_Presence | 1.044 | 0.997 |
|  | (1.037, 1.050) | (0.982, 1.012) |
|  | $p = 0.000^*$ | $p = 0.684$ |
| Constant | 49,784,537.000 | 489,263.300 |
|  | (38,544,981.000, 64,349,935.000) | (181,481.600, 1,331,364.000) |
|  | $p = 0.000^*$ | $p = 0.000^*$ |
| Observations | 6,155 | 811 |
| Log Likelihood | −56,988.530 | −2,805.794 |
| Akaike Inf. Crit. | 113,983.100 | 5,617.588 |
| Note: | | $^*p<0.05$ |

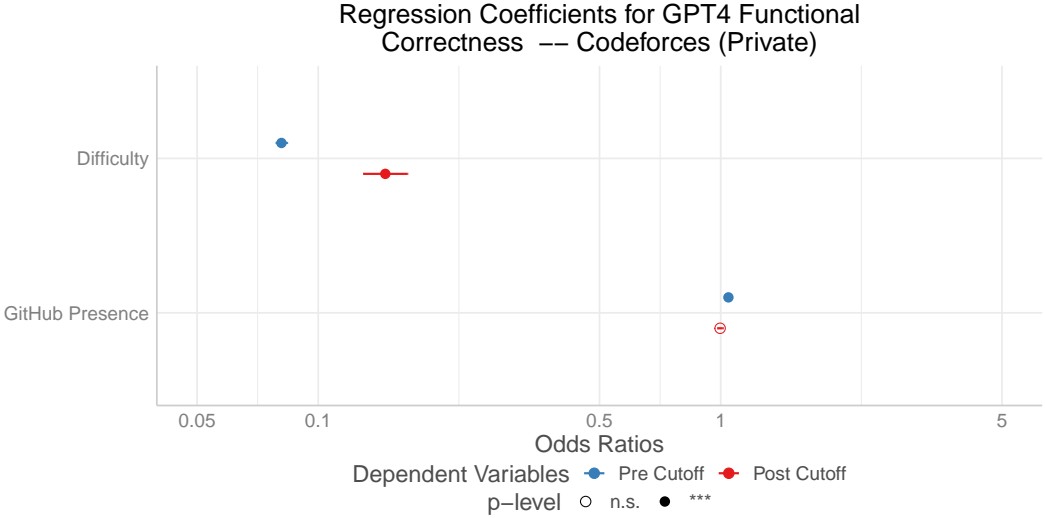

Figure 21: Regression coefficients plots of Pass Rate for GPT-4 on the Project Codeforces Dataset (evaluated on private test cases only). See Table 7 for regression coefficients.

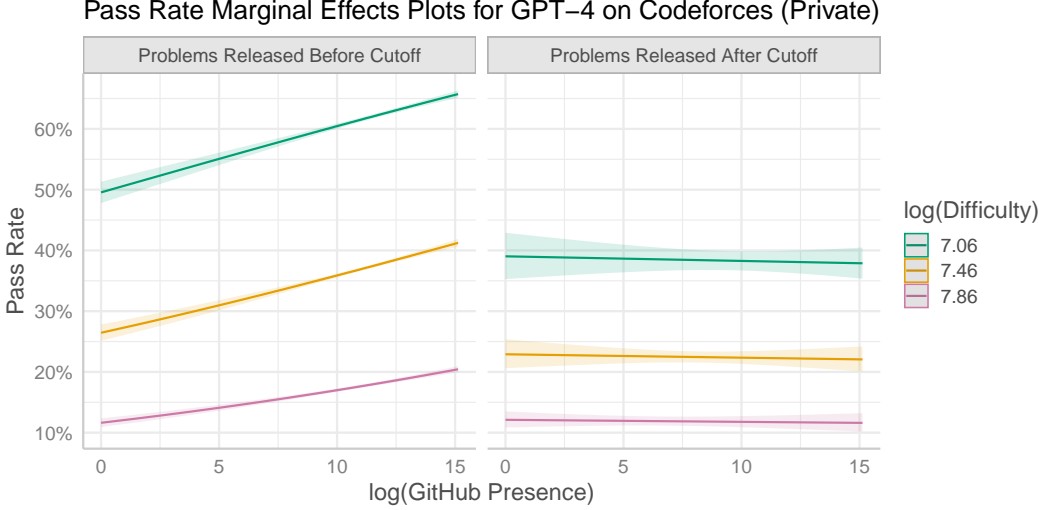

Figure 22: Marginal Effects of Pass Rate for GPT-4 on the Codeforces Dataset (evaluated on private test cases only)

Table 8: Regression table for Pass Rate of GPT-3.5-Turbo on the Codeforces dataset (private test cases only). See Figure 21 for plot of regression coefficients.

|  | *Dependent variable:* | |
| --- | --- | --- |
|  | Pass Rate (Private) | |
|  | Before Cutoff | After Cutoff |
|  | (1) | (2) |
| Difficulty | 0.112 | 0.168 |
|  | (0.109, 0.116) | (0.147, 0.192) |
|  | p = 0.000* | p = 0.000* |
| GitHub_Presence | 1.023 | 1.014 |
|  | (1.017, 1.030) | (0.996, 1.031) |
|  | p = 0.000* | p = 0.128 |
| Constant | 3,541,250.000 | 107,436.200 |
|  | (2,746,064.000, 4,569,487.000) | (37,418.200, 311,168.500) |
|  | p = 0.000* | p = 0.000* |
| Observations | 6,154 | 811 |
| Log Likelihood | −49,348.890 | −2,208.268 |
| Akaike Inf. Crit. | 98,703.780 | 4,422.537 |
| *Note:* | | *p<0.05 |

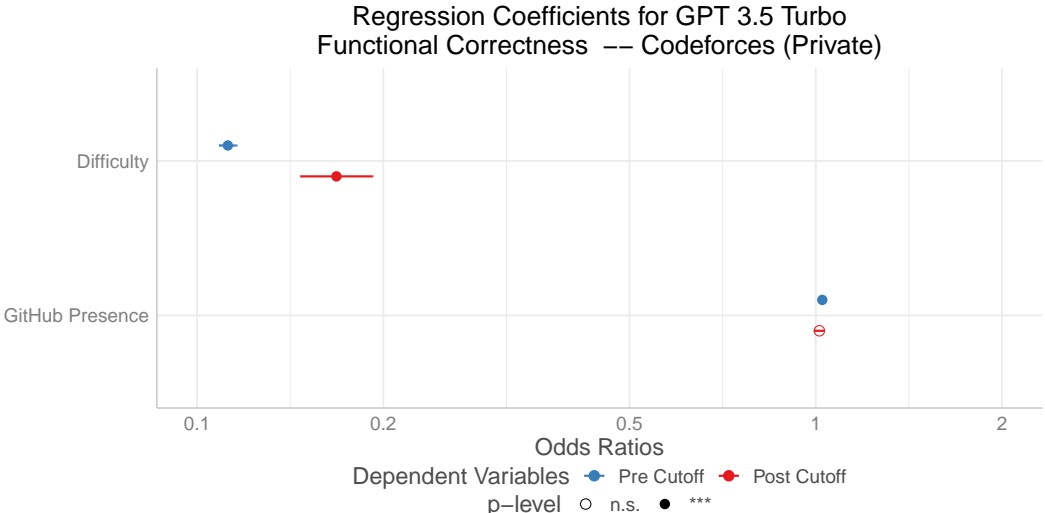

Figure 23: Regression coefficients plots of Pass Rate for GPT-3.5-Turbo on the Project Codeforces Dataset (evaluated on public test cases only). See Table 8 for regression coefficients.

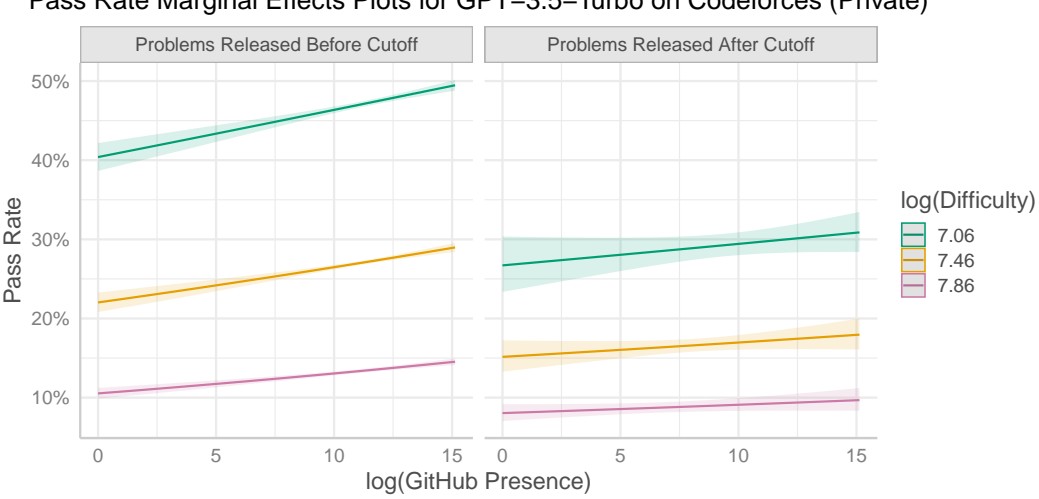

Figure 24: Marginal Effects of Pass Rate for GPT-3.5-Turbo on the Codeforces Dataset (evaluated on private test cases only)

Table 9: Code Bison Functional Correctness – Codeforces (Private)

| | Dependent variable: | |
| --- | --- | --- |
| | Pass Rate | |
| | Before Cutoff | After Cutoff |
| | (1) | (2) |
| Difficulty | 0.215 | 0.305 |
| | (0.208, 0.222) | (0.210, 0.442) |
| | p = 0.000* | p = 0.000* |
| | | |
| Github_Presence | 1.031 | 1.040 |
| | (1.025, 1.038) | (0.992, 1.091) |
| | p = 0.000* | p = 0.111 |
| | | |
| Constant | 15,512.470 | 574.153 |
| | (11,955.840, 20,133.940) | (32.974, 10,079.970) |
| | p = 0.000* | p = 0.00002* |
| | | |
| Observations | 6,791 | 132 |
| Log Likelihood | −36,805.650 | −211.771 |
| Akaike Inf. Crit. | 73,617.300 | 429.541 |

*Note:*                *p<0.05; **p<[0.**]; ***p<[0.***]

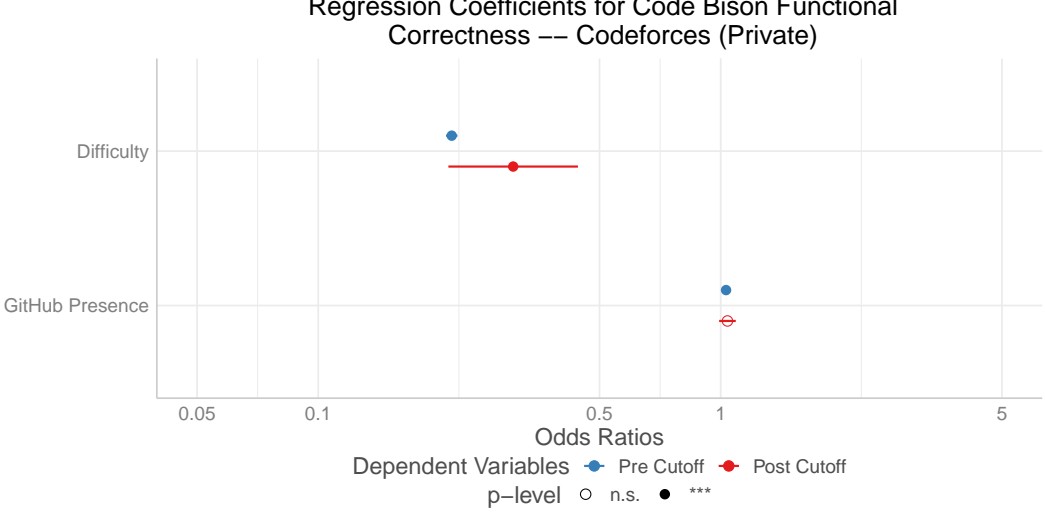

Figure 25: Regression coefficients plots of Pass Rate for Code Bison on the Project Codeforces Dataset (evaluated on public test cases only). See Table 9 for regression coefficients.

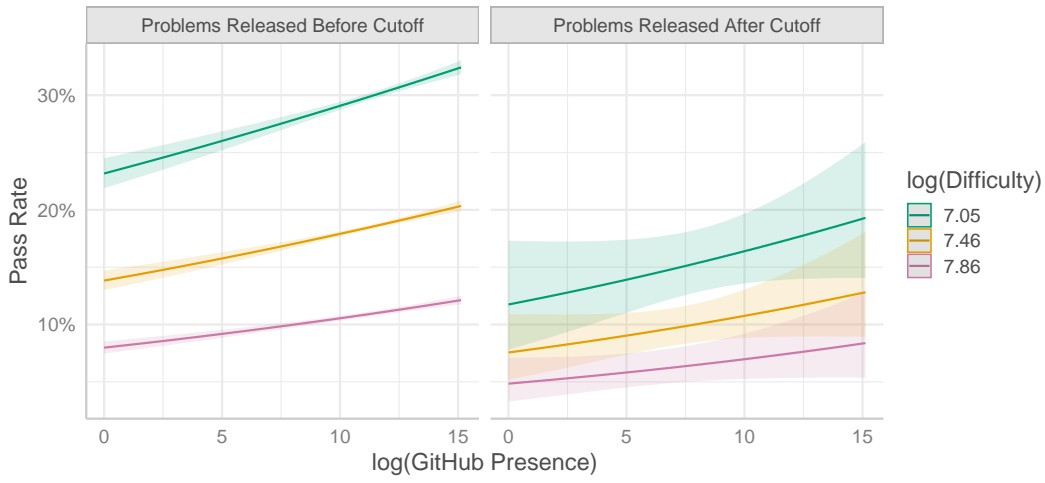

Figure 26: Marginal Effects of Pass Rate for Code Bison on the Codeforces Dataset (evaluated on private test cases only)

Table 10: Regression table for Pass Rate of GPT4 on the Project Euler dataset. See Figure 27 for plot of regression coefficients.

|  | Dependent variable: | |
| --- | --- | --- |
|  | Pass Rate | |
|  | Before Cutoff | After Cutoff |
|  | (1) | (2) |
| Difficulty | 0.145 | 1.000 |
|  | (0.104, 0.196) | (0.000, Inf.000) |
|  | p = 0.000* | p = 1.000 |
|  |  |  |
| GitHub_Presence | 1.476 | 1.000 |
|  | (1.127, 1.947) | (0.000, Inf.000) |
|  | p = 0.005* | p = 1.000 |
|  |  |  |
| Constant | 0.323 | 0.000 |
|  | (0.007, 15.405) | (0.000, Inf.000) |
|  | p = 0.567 | p = 1.000 |
|  |  |  |
| Observations | 765 | 72 |
| Log Likelihood | −170.511 | −0.000 |
| Akaike Inf. Crit. | 347.023 | 6.000 |
| *Note:* | *p<0.05; **p<[0.**]; ***p<[0.***] | |

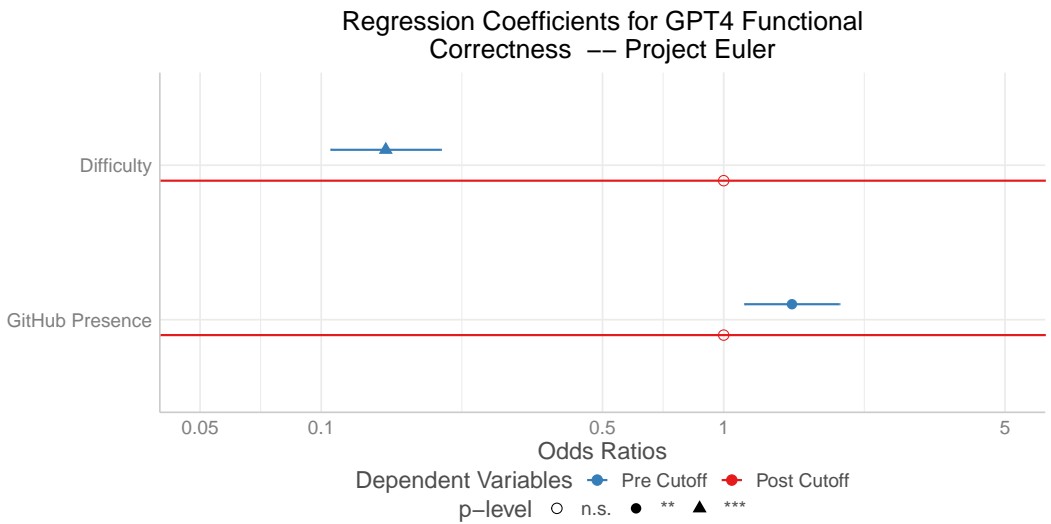

Figure 27: Regression coefficients plots of Pass Rate for GPT-4 on the Project Euler Dataset. See Table 10 for regression coefficients.

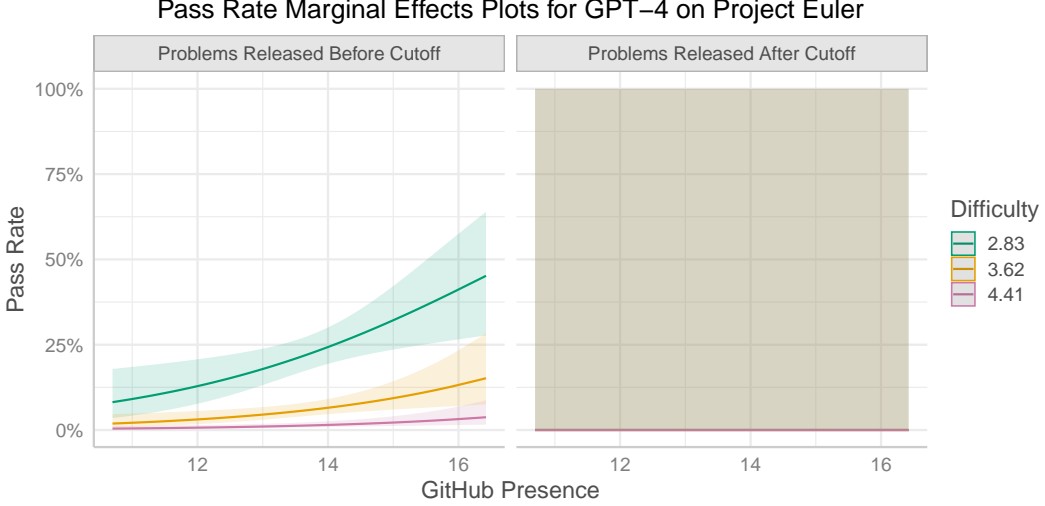

Figure 28: Marginal Effects of Pass Rate for GPT-4 on the Project Euler Dataset

Table 11: Regression table for Pass Rate of GPT-3.5-Turbo on the Project Euler dataset. See Figure 29 for plot of regression coefficients.

| | *Dependent variable:* | |
| --- | --- | --- |
| | Pass Rate | |
| | Before Cutoff | After Cutoff |
| | (1) | (2) |
| Difficulty | 0.166 | 1.000 |
| | (0.117, 0.228) | (0.000, Inf.000) |
| | p = 0.000* | p = 1.000 |
| | | |
| GitHub_Presence | 1.274 | 1.000 |
| | (0.954, 1.714) | (0.000, Inf.000) |
| | p = 0.100 | p = 1.000 |
| | | |
| Constant | 0.839 | 0.000 |
| | (0.012, 56.905) | (0.000, Inf.000) |
| | p = 0.936 | p = 1.000 |
| | | |
| Observations | 765 | 72 |
| Log Likelihood | −140.542 | −0.000 |
| Akaike Inf. Crit. | 287.084 | 6.000 |
| *Note:* | *p<0.05; **p<[0.**]; ***p<[0.***] | |

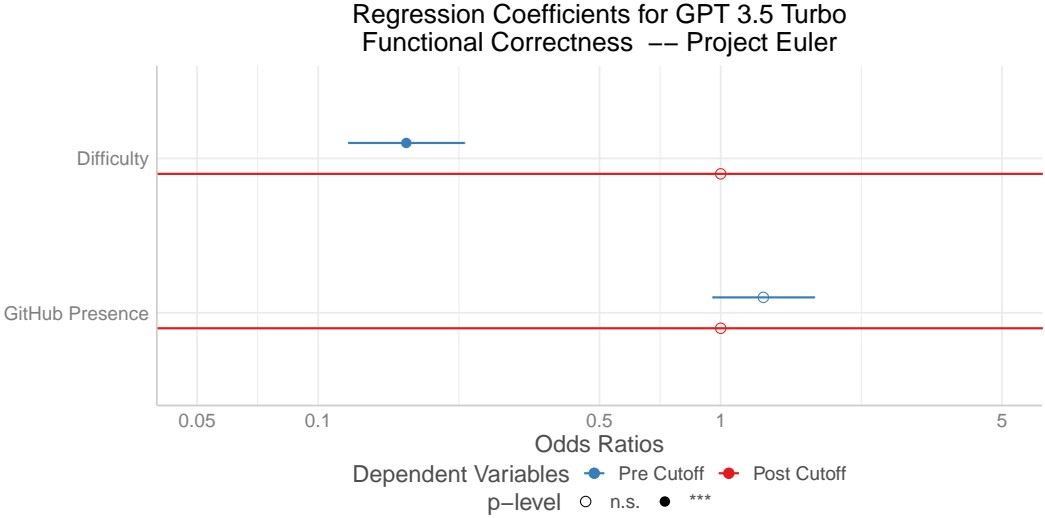

Figure 29: Regression coefficients plots of Pass Rate for GPT-3.5-Turbo on the Project Euler Dataset. See Table 11 for regression coefficients.

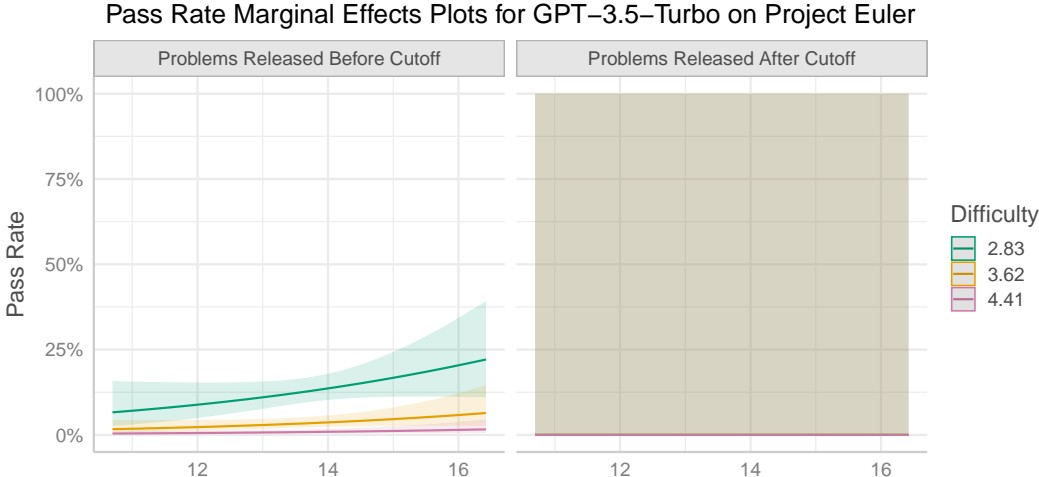

Figure 30: Marginal Effects of Pass Rate for GPT-3.5-Turbo on the Project Euler Dataset

Table 12: Regression table for Title Reproduction of GPT-4 on the Codeforces dataset. See Figure 31 for plot of regression coefficients.

| | Dependent variable: | |
| --- | --- | --- |
| | Title Reproduction | |
| | Before Cutoff | After Cutoff |
| | (1) | (2) |
| Difficulty | 0.897 | 0.898 |
| | (0.826, 0.974) | (0.756, 1.067) |
| | p = 0.010* | p = 0.222 |
| | | |
| GitHub_Presence | 1.010 | 1.005 |
| | (0.994, 1.026) | (0.982, 1.028) |
| | p = 0.210 | p = 0.684 |
| | | |
| Constant | 0.817 | 0.750 |
| | (0.418, 1.598) | (0.191, 2.956) |
| | p = 0.556 | p = 0.682 |
| | | |
| Observations | 18,446 | 3,954 |
| Log Likelihood | −11,074.250 | −2,265.482 |
| Akaike Inf. Crit. | 22,154.490 | 4,536.963 |
| *Note:* | | *p<0.05 |

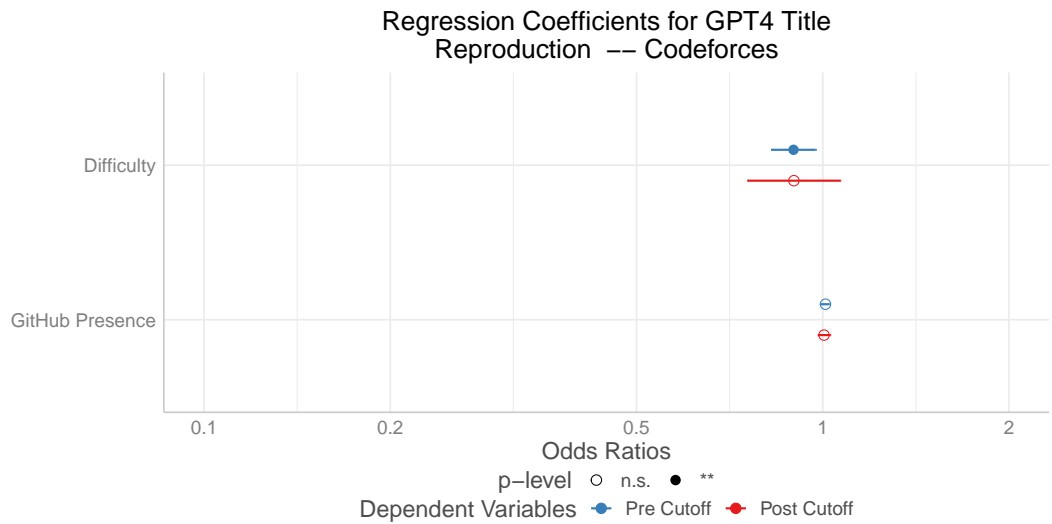

Figure 31: Regression coefficients plots of Title Reproduction Metric for GPT-4 on the Project Codeforces Dataset. See Table 12 for regression coefficients.

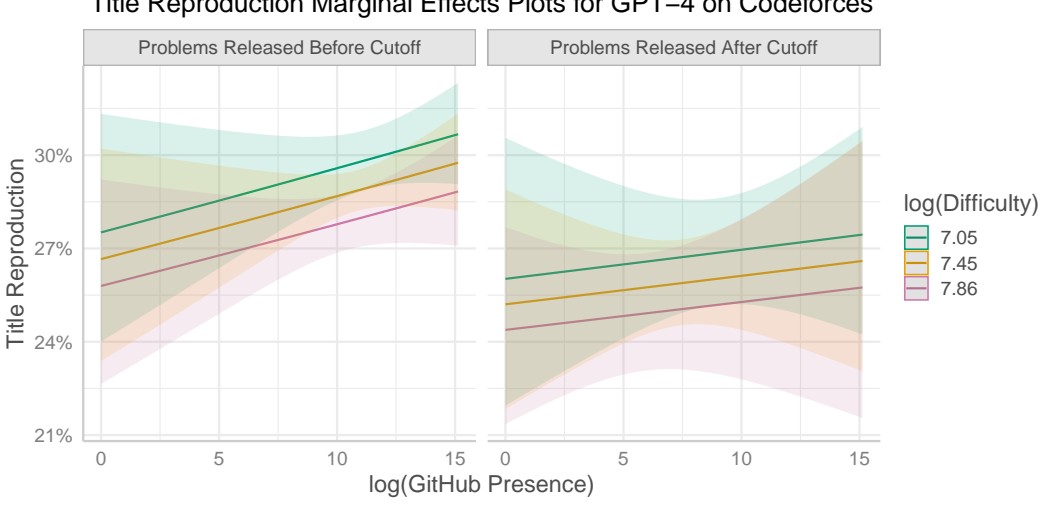

Figure 32: Marginal Effects of Title Reproduction Metric for GPT-4 on the Codeforces Dataset

Table 13: Regression table for Title Reproduction of GPT-3.5-Turbo on the Codeforces dataset. See Figure 33 for plot of regression coefficients.

| | Dependent variable: | |
| --- | --- | --- |
| | Title Reproduction | |
| | Before Cutoff | After Cutoff |
| | (1) | (2) |
| Difficulty | 0.839 | 0.709 |
| | (0.719, 0.979) | (0.510, 0.984) |
| | p = 0.026* | p = 0.041* |
| | | |
| GitHub_Presence | 1.010 | 0.973 |
| | (0.981, 1.041) | (0.931, 1.016) |
| | p = 0.506 | p = 0.216 |
| | | |
| Constant | 0.220 | 0.954 |
| | (0.063, 0.764) | (0.071, 13.105) |
| | p = 0.018* | p = 0.972 |
| | | |
| Observations | 18,446 | 3,954 |
| Log Likelihood | −4,293.516 | −869.667 |
| Akaike Inf. Crit. | 8,593.032 | 1,745.333 |
| Note: | | *p<0.05 |

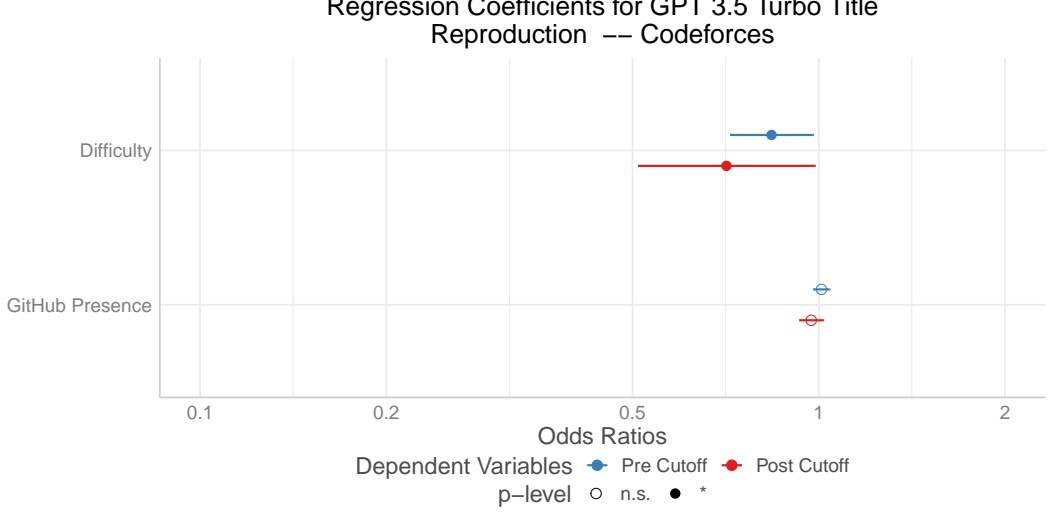

Figure 33: Regression coefficients plots of Title Reproduction Metric for GPT-3.5-Turbo on the Project Codeforces Dataset. See Table 13 for regression coefficients.

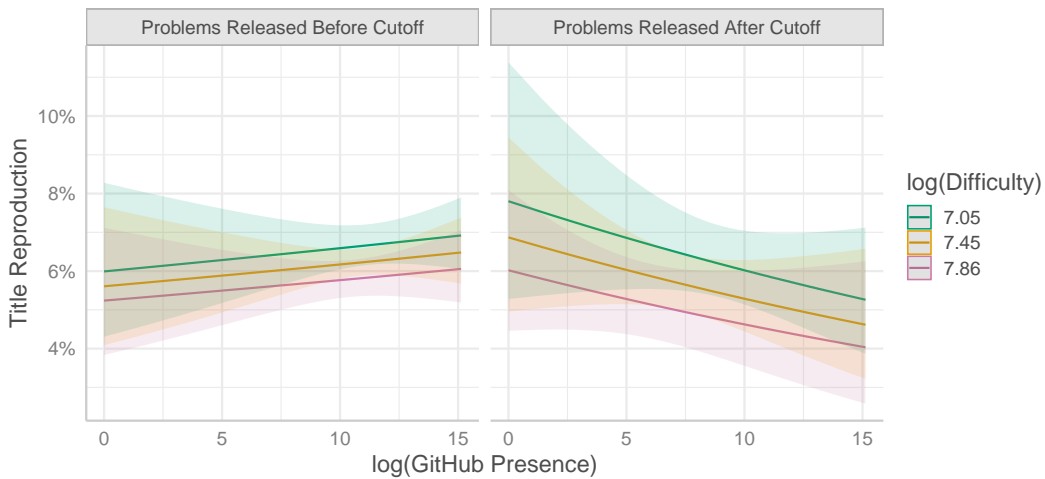

Figure 34: Marginal Effects of Title Reproduction Metric for GPT-3.5-Turbo on the Codeforces Dataset

Table 14: Regression table for Title Reproduction of GPT4 on the Project Euler dataset. ee Figure 35 for plot of regression coefficients.

| | *Dependent variable:* | |
| --- | --- | --- |
| | Title Reproduction | |
| | Before Cutoff | After Cutoff |
| | (1) | (2) |
| Difficulty | 0.780 | 0.946 |
| | (0.699, 0.870) | (0.631, 1.438) |
| | p = 0.00001* | p = 0.793 |
| | | |
| GitHub_Presence | 1.014 | 0.998 |
| | (0.962, 1.073) | (0.741, 1.344) |
| | p = 0.608 | p = 0.991 |
| | | |
| Constant | 0.848 | 0.508 |
| | (0.349, 1.980) | (0.008, 29.457) |
| | p = 0.709 | p = 0.745 |
| | | |
| Observations | 2,556 | 190 |
| Log Likelihood | −1,536.829 | −114.284 |
| Akaike Inf. Crit. | 3,079.658 | 234.568 |
| *Note:* | *p<0.05; **p<[0.**]; ***p<[0.***] | |

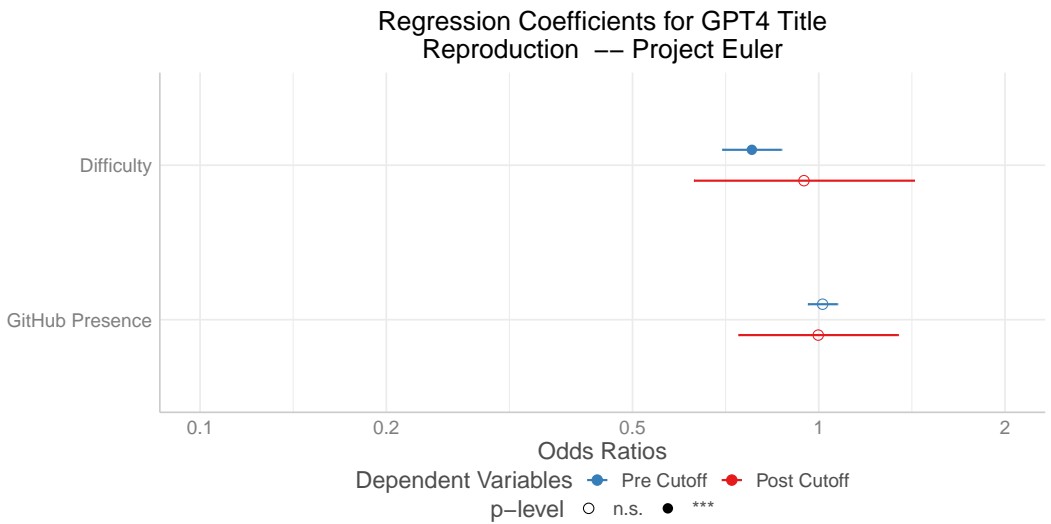

Figure 35: Regression coefficients plots of Title Reproduction Metric for GPT-4 on the Project Euler Dataset. See Table 13 for regression coefficients.

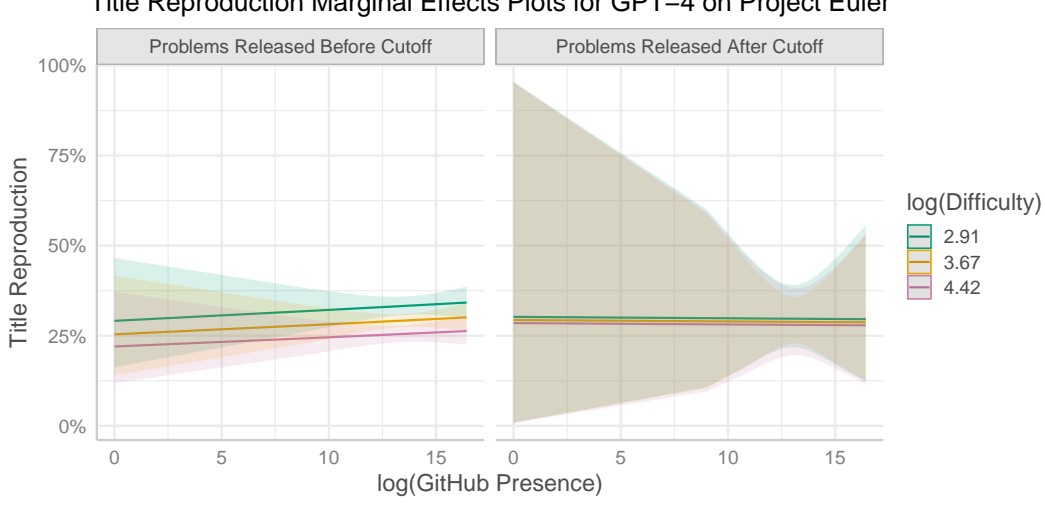

Figure 36: Marginal Effects of Title Reproduction Metric for GPT-4 on the Project Euler Dataset

Table 15: Regression table for Title Reproduction of GPT-3.5-Turbo on the Project Euler dataset. ee Figure 37 for plot of regression coefficients.

| | Dependent variable: | |
| --- | --- | --- |
| | Title Reproduction | |
| | Before Cutoff | After Cutoff |
| | (1) | (2) |
| Difficulty | 1.178 | 0.643 |
| | (0.934, 1.517) | (0.136, 4.339) |
| | p = 0.184 | p = 0.589 |
| GitHub_Presence | 0.981 | 1.283 |
| | (0.898, 1.094) | (0.339, 6.026) |
| | p = 0.696 | p = 0.715 |
| Constant | 0.044 | 0.002 |
| | (0.007, 0.198) | (0.000, 51,280.610) |
| | p = 0.0002* | p = 0.505 |
| Observations | 2,556 | 190 |
| Log Likelihood | −564.246 | −10.900 |
| Akaike Inf. Crit. | 1,134.492 | 27.801 |
| *Note:* | *p<0.05; **p<[0.**]; ***p<[0.***] | |

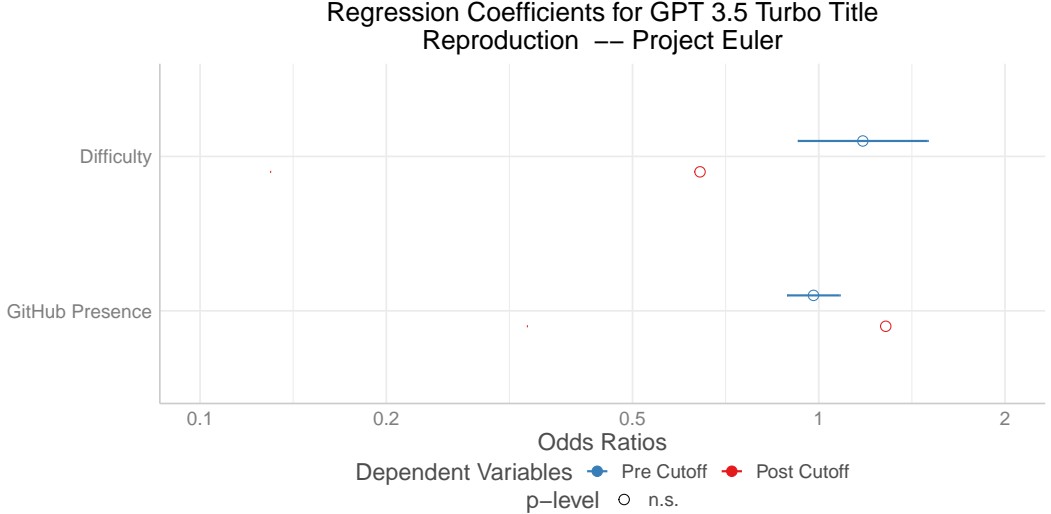

Figure 37: Regression coefficients plots of Title Reproduction Metric for GPT-3.5-Turbo on the Project Euler Dataset. See Table 13 for regression coefficients.

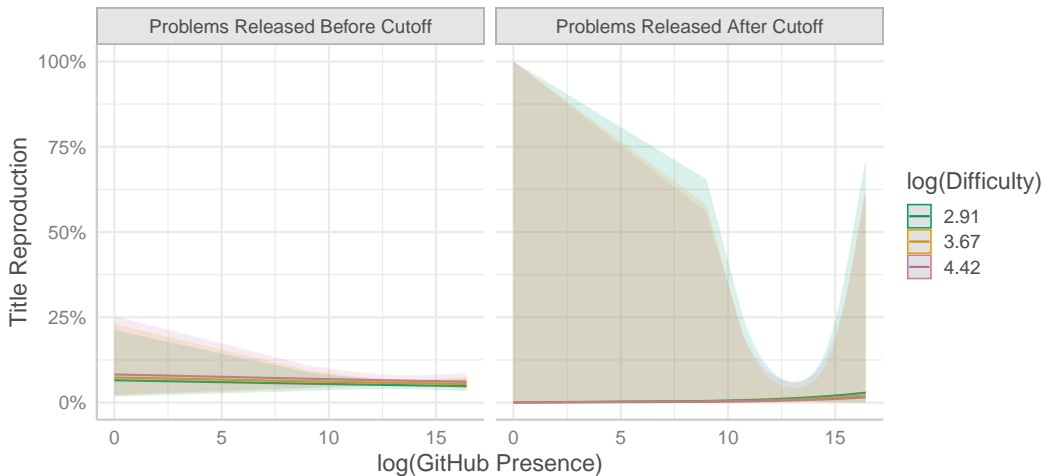

Figure 38: Marginal Effects of Title Reproduction Metric for GPT-3.5-Turbo on the Project Euler Dataset

Table 16: Regression table for Tag Reproduction of GPT-4 on the Codeforces dataset. See Figure 39 for plot of regression coefficients.

|  | *Dependent variable:* | |
| --- | --- | --- |
|  | Tags Reproduction | |
|  | Before Cutoff | After Cutoff |
|  | (1) | (2) |
| Difficulty | 0.431 | 0.826 |
|  | (0.400, 0.465) | (0.707, 0.966) |
|  | p = 0.000* | p = 0.017* |
|  |  |  |
| GitHub_Presence | 0.991 | 0.999 |
|  | (0.977, 1.005) | (0.981, 1.017) |
|  | p = 0.195 | p = 0.888 |
|  |  |  |
| Constant | 310.696 | 1.026 |
|  | (168.396, 573.834) | (0.299, 3.508) |
|  | p = 0.000* | p = 0.968 |
|  |  |  |
| Observations | 24,474 | 6,425 |
| Log Likelihood | −15,399.190 | −3,172.626 |
| Akaike Inf. Crit. | 30,804.370 | 6,351.252 |
| *Note:* |  | *p<0.05 |

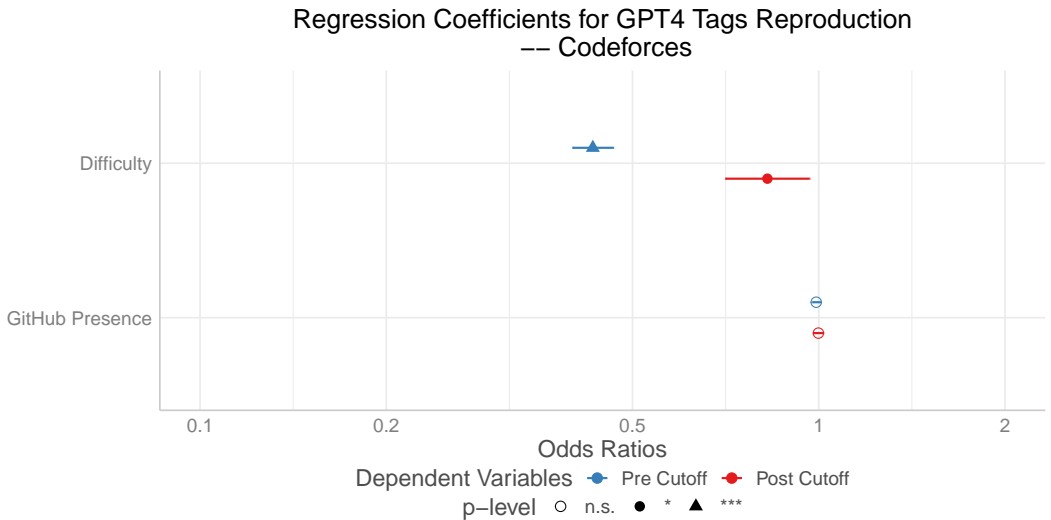

Figure 39: Regression coefficients plots of Tag Reproduction Metric for GPT-4 on the Project Codeforces Dataset. See Table 16 for regression coefficients.

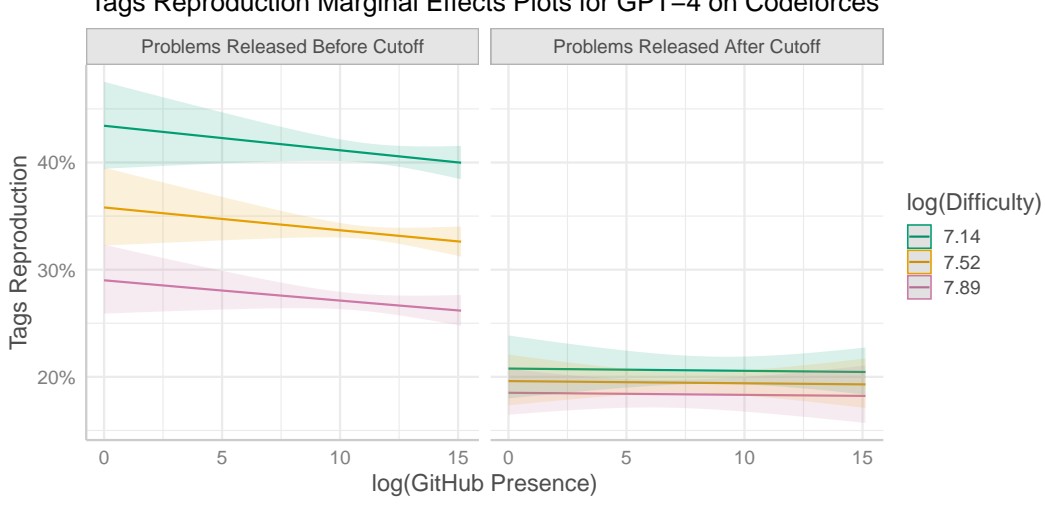

Figure 40: Marginal Effects of Tag Reproduction Metric for GPT-4 on the Codeforces Dataset

Table 17: Regression table for Tag Reproduction of GPT-3.5-Turbo on the Codeforces dataset. See Figure 41 for plot of regression coefficients.

| | Dependent variable: | |
|---|---|---|
| | Tags Reproduction | |
| | Before Cutoff | After Cutoff |
| | (1) | (2) |
| Difficulty | 0.497 | 0.739 |
| | (0.452, 0.547) | (0.619, 0.883) |
| | p = 0.000* | p = 0.001* |
| | | |
| GitHub_Presence | 0.959 | 1.001 |
| | (0.942, 0.976) | (0.980, 1.022) |
| | p = 0.00001* | p = 0.955 |
| | | |
| Constant | 56.613 | 1.540 |
| | (26.265, 121.894) | (0.380, 6.228) |
| | p = 0.000* | p = 0.545 |
| | | |
| Observations | 24,474 | 6,425 |
| Log Likelihood | −10,635.220 | −2,587.031 |
| Akaike Inf. Crit. | 21,276.450 | 5,180.061 |
| Note: | | *p<0.05 |

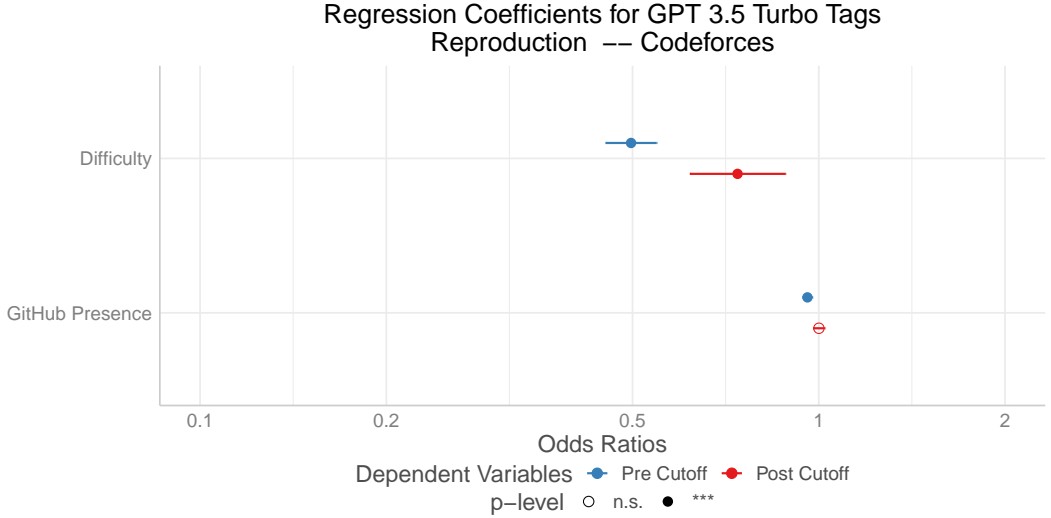

Figure 41: Regression coefficients plots of Tag Reproduction Metric for GPT-3.5-Turbo on the Project Codeforces Dataset. See Table 17 for regression coefficients.

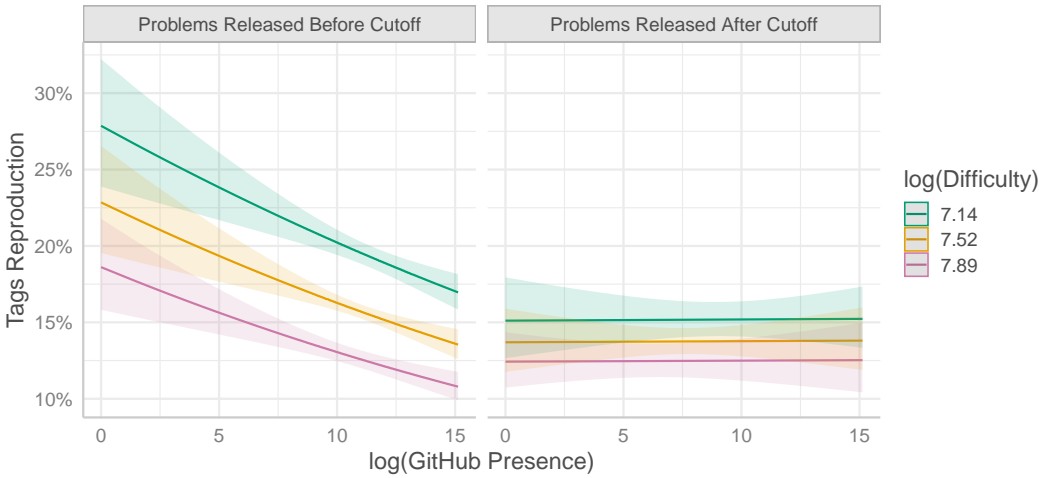

Figure 42: Marginal Effects of tag Reproduction Metric for GPT-3.5-Turbo on the Codeforces Dataset

