# OpenReview forum: "To the Cutoff... and Beyond? A Longitudinal Perspective on LLM Data Contamination"
_ICLR.cc/2024/Conference — ICLR 2024 poster_

### Official Review · Reviewer_yfjE · 2023-10-22

**Soundness:** 3 good
**Presentation:** 3 good
**Contribution:** 2 fair
**Rating:** 8
**Confidence:** 4

**Summary:**

The paper investigates data contamination of GPT-3.5-Turbo and GPT-4 with problems from Codeforces and Project Euler. It does so by analysing the passrates in relation to the GitHub presence and notes a positive correlation for data before the cutoff, but no significant correlation after this date.

**Strengths:**

1. Identifying data contamination is an important issue, especially for evaluation datasets that are often used to create rankings.
2. Including problem difficulty as an independent variable is an important step in isolating the confounding effect of item difficulty on pass rates.
3. I appreciate the openness in referencing blog posts and tweets that anecdotally suggested possible contamination prior to this work

**Weaknesses:**

1. The methodology is only applied to GPT-3.5/GPT-4, where training details are unknown. In particular, as noted in footnote 1, OpenAI has admitted to using a small amount of data beyond the cutoff date. While I understand the choice of the GPT family as a commonly used model, it would have been better to verify the approach with fully open models where more training details are available (and more trustworthy).
2. The methodology requires underlying datasets that are longitudinal in nature, i.e. release problems/individual tasks over time; this limits the applicability to sources other than Project Euler / Codeforces.

**Questions:**

### Minor Comments
* Particularly in section 2, some citations are formatted differently, with the author names outside the parentheses; in sequences of different citations, readability could be improved by using the same citation format as in section 1.

---

> ### Author Response · Authors · 2023-11-16
> **Reply to Reviewer yfjE**
>
> We thank you very much for your thorough and extensive review. We appreciate that you find our work tackles an important issue, while controlling for confounding variables. We reply to your questions below.
>
> **1. Additional LLMs.**
> Thank you for the suggestion! We originally focused on GPT-3.5 and GPT-4 for a few critical reasons: (i) relevance to the large community, in and out of research, that rely on these models, (ii) these models’ high performance on code generation tasks. With respect to this second point, we clarify that when a model’s pass rate on these problems is prohibitively low, we are effectively unable to observe non-trivial trends or extract longitudinal insights–and we have found this to be the case for some additional models we have tested. For example, GPT-3.5 achieved 27% pass rate before the cutoff and 13% after,  while GPT-4 achieved 37% pass rate before and 16% after. In contrast, the pass rates we obtained for Text-Davinci-002 with a comparable prompting strategy are less than 1% before *and* after–yielding it impossible to conduct a substantive analysis featuring functional correctness.
>
> We also are actively collecting data from some other models, including a modern open-source LLM, and plan on providing these results when the results are available soon (we aim to do so before the end of this discussion window). This discussion is also repeated in the new Appendix B.6.
>
> **2. Longitudinal datasets.**
>
> We agree that our analysis requires longitudinal datasets; however, we anticipate that the popularity of such benchmarks will continue to rise, as releasing datasets over time is potentially the only foolproof method for avoiding data contamination. For example, creators of recent benchmarks such as [BIG-Bench](https://arxiv.org/abs/2206.04615) are consciously updating their benchmarks over time. Even other non-LLM benchmarks such as [OpenML Benchmarking Suites](https://arxiv.org/abs/1708.03731) are being updated over time, since there is a related push to ensure that the community [does not overfit](https://arxiv.org/abs/2112.01716) to any single set of benchmarks. We are optimistic that as staggered release of benchmark datasets becomes increasingly common and/or an accepted best practice, that the framework we present here can be more widely applied.
>
> Furthermore, we note that while our analysis can only be conducted on longitudinal datasets, its implications extend to any dataset which can be contained in a webscale training dataset; it leverages longitudinal structure in a test benchmark to reveal the massive contamination effect which can be observed on “seen” examples of any dataset.
>
> **3. Minor comments.**
> Thank you for catching these formatting issues; they have now been fixed in the updated pdf.
>
> Thank you once again, for your excellent points. Please let us know if you have any follow-up questions or further suggestions. We are still conducting new analyses and are excited to update you with their results. We would be happy to continue the discussion.

---

> ### Author Response · Authors · 2023-11-22
> **New Models**
>
> Thank you again for your thoughtful review. We are writing with more experiments which we have conducted during the rebuttal period which address you concern *W1: Additional LLMs*. We liked your suggestion of adding more LLMs beyond the GPT-4, GPT-3.5, and Davinci models which were in our original manuscript.
>
> We have now added the analysis of two more LLMs on Codeforces: [Codey/Code Bison](https://cloud.google.com/vertex-ai/docs/generative-ai/code/code-models-overview) (Google’s code generation foundation model, released around the same time as PaLM 2), and [Code-Llama](https://huggingface.co/codellama).
>
> As for Code Bison, we very interestingly observe the same behavior that we did for GPT-4 and GPT-3.5-Turbo. Specifically, after the training cutoff of Code Bison, we see that the GitHub presence metric is no longer a significant predictor of functional correctness for this model. code-bison@001’s training cutoff, although not publicly known, is probably around February 2023, as this model was released [within weeks](https://cloud.google.com/vertex-ai/docs/generative-ai/learn/model-versioning) of text-bison@001 and chat-bison@001, which have [known training cutoffs of February 2023](https://cloud.google.com/vertex-ai/docs/generative-ai/learn/models). Accordingly, we conducted our analysis assuming this February 2023 cutoff. We have updated our manuscript to show this by updating Figures 2 and including all additional figures and tables in the appendix as with the GPT models.
>
> The results from this additional model shows that the training cutoff itself is a moderating factor in the code generation completion performance. This conclusion is further bolstered by the almost 2 year separation between the GPT cutoff and the Code Bison cutoff and yet still producing the same qualitative finding.
>
> We additionally worked with Code-Llama and attempted to the best of our abilities to get it to perform on these questions. Unfortunately, even on the largest and most capable model for our use case, 34b-Instruct, we could not get Code-Llama to output answers which yielded pass rates above 1% on Codeforces. This result follows that of our observation around Text-Davinci-002.
>
> We hope that these additional experiments address your main concern about the limited number and models in our experiments. We now conduct analysis of 5 LLMs with various cutoff dates, and we observe signs of contamination for those models which yield pass rates above 1%. We look forward to answering any additional questions you may have, before the author response period closes tomorrow (end of day Nov 22nd). Thanks!

---

> > ### Comment · Reviewer_yfjE · 2023-11-22
> >
> > Thank you for your comments and updated results. I will update my score accordingly.
> >
> > Just as a remark:
> >
> > > We additionally worked with Code-Llama and attempted to the best of our abilities to get it to perform on these questions. Unfortunately, even on the largest and most capable model for our use case, 34b-Instruct, we could not get Code-Llama to output answers which yielded pass rates above 1% on Codeforces. This result follows that of our observation around Text-Davinci-002.
> >
> > I suspect this may be an artifact of "pass rate" being a discontinuous metric (on a single sample) that cannot adequately measure *partial* progress. It would be interesting to look for a smoother metric. A simple choice might be to look at the perplexity of a/the correct solution, but I am not sure how well that would be calibrated.

---

### Official Review · Reviewer_EfiM · 2023-10-30

**Soundness:** 3 good
**Presentation:** 4 excellent
**Contribution:** 3 good
**Rating:** 8
**Confidence:** 4

**Summary:**

Assess whether GPT performance at coding (sometimes called program synthesis) was possibly affected by contamination of pretraining data using a naturally occurring experiment (i.e. comparing scores before and after the pretraining knowledge cutoff dates).

**Strengths:**

Overall, I really liked this paper. I thought it was well motivated, clearly conceptualized, well executed and somewhat thorough. I have a small number of requested changes, and if the authors and I agree that the changes are sensible and if the authors agree to make the changes, I would be happy to increase my score.

**Weaknesses:**

> Figure 1: Marginal Effects of Pass Rate Metric

I think this is an amazing figure. 5 comments, ordered from minor to major:

1. easy: Stacking log(Github Presence) and log(Difficulty) at the bottom makes reading the figure tricky. I might suggest moving log(Difficulty) to the right side.

2. easy: GitHub is stylized "GitHub", not "Github"

3. medium: Where is the equivalent plot for Project Euler? I might have missed this, but I cannot find it in the main text or appendix.

4. hard: The pass rate is significantly lower for easy and medium problems, even for log(Github Presence) = 0. I understand that GitHub Presence is a proxy, but I would think that log(GitHub Presence) = 0 is our best guess for "low or no contamination", but there's still a 10-20% decrease in pass rate. Why? I can think of 2-4 possible answers: (a) GPT-4 genuinely becomes much worse after the knowledge cutoff; (b) GitHub presence is inadequate and/or misleading, (c) the distribution of Codeforce problems changed after GPT-4 was finished pretraining, or (d) something changed in how the pass rate is calculated on generated outputs. More explanations might also be possible. Is there some way for the authors to try to investigate the cause of this shift?

5. hard: I was hoping for either a qualitative or quantitative analysis about what GPT-4 is outputting on Codeforces problems released after the cutoff, but I can't find even a single example of the raw generated outputs. Could the authors please provide some manual examples, even in the appendix, to convincingly demonstrate that GTP-4 is indeed outputting worse code? I want to rule out that silly possibilities (e.g., a shift in formatting) are affecting the results.

> Table 1

I personally find Tables are less effective at communicating than Figures. Since these are regression tables, could you possibly consider switching to a Forest plot of regression coefficients? Some random examples here:

- https://www.researchgate.net/figure/Forest-plot-of-regression-coefficients-95-confidence-interval-for-the-association_fig1_331119872
- https://www.researchgate.net/figure/Coefficient-plots-from-linear-regression-predicting-what-makes-an-interaction-meaningful_fig1_343608677
- http://www.strengejacke.de/sjPlot/reference/plot_models.html.

To make my suggestion as concrete as possible, using terminology from matplotlib & seaborn (assuming you're using Python, but I'm sure R could do this as well), I'm specifically thinking that your X axis should be the estimated parameters and confidence intervals, Y would be the covariates (i.e. Difficulty & GitHub presence), the Hue is either Before Cutoff or After Cutoff, and you have two side-by-side axes, one for GPT4 and the other for GPT3.5.

I personally would prefer all regression tables to be visualized as such (Tables 1, 2, and those in the appendix).

**Questions:**

Not a question, but I want to note that:

1. I like the use of Pass Rate in lieu of pass@1. I think that's a very sensible choice.

2. I like the citation of Horace He's and Chris Cundy's tweets. Very good scholarship, even if Tweets aren't "published" in a traditional sense.

---

> ### Author Response · Authors · 2023-11-16
> **Reply to Reviewer EfiM (1/2)**
>
> We thank you very much for your thorough and enthusiastic review. We appreciate that you find our work well motivated, clearly conceptualized, and well-executed. We agree with all of your suggestions and have done our best to address them below in and in the updated manuscript. Please let us know if we have addressed your comments.
>
> **W1+2: Fig 1 minor changes.**
> Thank you for pointing out these stylistic changes. We have now moved the legend and updated “Github” -> “GitHub”.
>
> **W3: Fig 1 for Project Euler.**
> Thank you; this plot can be found in Figure 17; the associated plot for GPT3.5 can be found in Figure 19. Here you’ll see how the performance of these systems on Project Euler is much decreased on problems released before the training cutoffs, and entirely wiped out for problems released after the cutoff — neither GPT-4 or GPT-3.5 yield any pass rates above 0 for problems after the training cutoff.
>
> **W4: Pass rate when log(Github Presence)=0.**
> This is a great insight! We believe that the main factor at play is b) - GitHub Presence is a strong indicator of availability in the training set, but it still underestimates the true availability in the training set. For example, *all* Codeforces problems show up on the internet in several places (e.g., [here](https://cf.kira924age.com/#/table/), [here in pdf](https://github.com/AliOsm/PDF-CodeForces-Problems)). Therefore, we believe this is a major reason for why the completion rate is higher before the cutoff even for problems with log(GitHub Presence)=0.  We comment on your other hypotheses as well.
> 1. In terms of the models themselves, we used the exact same models when evaluating problems released before and after the cutoff date (in fact, the evaluations were performed in one large batch and later separated by date).
> 2. As we state above, we believe that this is the main factor. We agree that “GitHub presence” may under-estimate the total presence of a Codeforces problem in the GPT training data. For example, *all* Codeforces problems show up on the internet in several places (e.g., [here](https://cf.kira924age.com/#/table/), [here in pdf](https://github.com/AliOsm/PDF-CodeForces-Problems)).
> 3. It is challenging to fully rule out this possibility, but qualitatively, we cannot spot any change over time in the type of Codeforces problems released. It is possible that a difference in problems does cause a small change in pass rate after the cutoff, but we believe that the majority of the observed difference in pass rate can be attributed to (2).
>     1. To investigate, we conduct a set of additional experiments to assess whether the distribution over tags (only available for Codeforces) and/or difficulty level (available for Codeforces and Project Euler) changed in a statistically significant way during the post-cutoff period, relative to the pre-cutoff period. We present this analysis in its entirety, consisting of qualitative plots and  $\chi^2$ tests, in Appendix B.7 of our updated submission pdf, and summarize key findings below:
>         - For **Codeforces**, we do not find any statistically significant difference in the distribution of normalized counts over problem tags during the pre-vs. post- period. We also do not find any statistically significant difference in the distribution over discretized difficulty scores during the pre-vs. Post-period.
>         - For **Project Euler**, problem tags are not publicly available, so we cannot perform tag analysis. We do not find any statistically significant difference in the distribution over discretized difficulty scores during the pre-vs. Post-period.
>         - These findings help to mitigate concerns that the drop-off in performance we observe might be attributable to significant changes in the distribution over tags (for Codeforces) and/or over difficulty levels during the post-cutoff period.
>
> 4. We ran the same evaluation script for all problems, and only then separated by date: https://anonymous.4open.science/r/to-the-cutoff-review-253A/eval/chronological_evaluation/chronological_dataset.py (line 162).
>
> (1/2)

---

> > ### Comment · Reviewer_EfiM · 2023-11-20
> > **Response to Authors (1/2)**
> >
> > Overall, I'm happy with your changes.
> >
> > > W1+2: Fig 1 minor changes.
> >
> > Wonderful - thank you!!
> >
> > > W3: Fig 1 for Project Euler. Thank you; this plot can be found in Figure 17; the associated plot for GPT3.5 can be found in Figure 19
> >
> > I think the figure numbering might have changed between your answer and the revised manuscript. I see Figure 10 has a caption "Figure 10: Marginal Effects of Pass Rate for GPT-4 on the Project Euler Dataset" but a title of "Functional Correctness Marginal Effects Plots for GPT−4 on Codeforces". Is this figure for Euler or Codeforces?
> >
> > > W4: Pass rate when log(Github Presence)=0. This is a great insight! We believe that the main factor at play is b) - GitHub Presence is a strong indicator of availability in the training set, but it still underestimates the true availability in the training set.
> >
> > This seems like a reasonable answer.

---

> ### Author Response · Authors · 2023-11-16
> **Reply to Reviewer EfiM (2/2)**
>
> **W5: GPT-generated outputs on problems released after the cutoff.**
>
> To address this question, we would like to: (a) make a minor methodological clarification; and (b) note that we have added a new section to the appendix (B.8) that contains numerous examples of GPT-4 and GPT-3.5.-Turbo output (i.e., generated code) for Codeforces problems released before and after the GPT training cutoff. (We restrict our attention to Codeforces because our Project Euler analysis focused on the correctness of generated numerical solutions, rather than evaluating the outputs of generated code).
>
> With respect to (a), we note for clarity that what we are trying to control for in our natural experiment are factors with the potential to influence an LLM’s ability to produce a functionally or numerically correct solution to a problem (i.e., the problem’s difficulty), and/or the likelihood that the LLM would have seen this problem during training (i.e., GitHub presence, since scraped GitHub repositories are part of the GPT training corpus). Thus, GPT need not produce *worse* code when we evaluate its ability to produce functionally correct solutions for problems released after the cutoff. Indeed, given that we perform all of our evaluations at a single point in time, the code-generating ability of each GPT instance is “fixed”--what is (potentially) changing is the artificially inflated performance benefit (or lack thereof) the model may demonstrate when it is evaluated on examples it *has* seen during training versus those it has not.
>
> With respect to (b)---i.e., our new appendix section—because the full description and generated output associated with a given problem can both be quite lengthy, we have constructed this new section by partitioning the Codeforces problems and along two dimensions: (1) release date pre- vs. post GPT cutoff; (2) discretized LLM functional correctness. With respect to (2), for a given LLM and problem, functional correctness is computed as the ratio of test cases that the LLM’s generated code passes (see Section 4 for details). Thus, functional correctness takes values in [0,1.0]. We discretize via the following mapping:
>
> $\lambda: x \mapsto 0 | (x \leq Q1) \lor 1 | (Q1 < x \leq Q2) \lor 2 | (Q2 < x \leq Q3) \lor 3 | x > Q3$
>
> where $x$ represents a given problem's raw difficulty score, and Q1, Q2, and Q3 correspond to the first, second, and third quartiles, respectively.
>
> We thus consider 8 subgroups per model (i.e., {pre, post} x {0,1,2,3}), and draw two examples uniformly at random from each subgroup. We present each generation along with corresponding problem title, ID, difficulty score, url, and the LLM’s functional correctness score. Our intent here is to avoid cherry-picking of results, while facilitating the visualization of a representative set of generations. We also note that the generated code produced by each model for *every* Codeforces problem is available in the results file that we provide as part of our supplementary material. We view additional/comparative static and behavioral analyses of these outputs as a promising direction for future work.
>
>
> **Table 1.**
> Thanks once again for this suggestion! We have now added these figures. We replaced Table 1 and Table 2 with their corresponding forest plots. Additionally, we have added the forest plots into the appendix for every regression coefficient table. These figures give new visual insights so the reader can quickly understand the relative relationship between the regression coefficients. For example, we can easily see that the coefficients for both Difficulty and GitHub presence get closer to 1 (the null effect) after the cutoff.
>
> Thank you once again, for your excellent points. Please let us know if you have any follow-up questions or further suggestions. We are still conducting new analyses and are excited to update you with their results. We look forward to continuing the discussion.
>
> (2/2)

---

> ### Comment · Reviewer_EfiM · 2023-11-20
> **Response to Authors (1/2)**
>
> > Figure 2
>
> I think this looks very nice. Thank you!! Same with Figure 3.
>
> > Thus, GPT need not produce _worse_ code when we evaluate its ability to produce functionally correct solutions for problems released after the cutoff.
>
> I'm not sure I understand this point. We might be using different terminology. I agree that the evaluations are run at a single point in time, and I believe your evidence (e.g., Figure 1) that increasing GitHub presence is correlated with functional correctness. What I was trying to confirm is that generated code for problems released after cutoff is "worse" in the sense that we would all agree the generated code for problems released after cutoff pass fewer test cases than generated code for problems released before cutoff. My intention was to rule out that the evaluation process itself somehow (potentially unintentionally) affected the performance.
>
> For example, suppose that before the cutoff, Codeforces required code to be submitted in format A (e.g., the main function should be called `main()`), but after the cutoff, Codeforces required code to be submitted in format B (e.g., the main function should be called `start()`). If so, I think it would be hardly surprising if a model pretrained on format A would perform less well when evaluated on format B, since the model wouldn't know that format A is deprecated and format B is appropriate?
>
> This is what I meant by confirming that the code post-cutoff is "wrong". Perhaps "less functionally useful" is a better term. I want to know that the generated code for problems post-cutoff is functionally less useful than code for problems pre-cutoff. I'm not interested in things like formatting, variable name choice, etc. that we might consider when discussing code quality.
>
> Could you please clarify what you meant by "GPT need not produce _worse_ code" post-cutoff? How is GPT-4 scoring worse post-cutoff if it isn't producing worse code?
>
>
>
> I went to increase your score to 7, but apparently 7 is not an option for ICLR. If we can reach agreement on this topic, and I think we can, then I would be happy to bump you up to an 8.

---

> > ### Author Response · Authors · 2023-11-22
> > **Second reply to Reviewer EfiM**
> >
> > Thanks for your reply; we are glad that you are overall happy with the updates!
> >
> > **Figure numbering.**
> > Yes, thank you for pointing this out. We have now corrected all plots and captions in the manuscript. The Project Euler plots are Fig 28 (GPT-4) and Fig 30 (GPT-3.5).
> >
> > **GPT need not produce worse code.**
> > Thank you for clarifying, and now we understand the confusion. Our statement “GPT need not produce worse code” was meant as a clarification to your original statement “[GPT-4] is indeed outputting worse code [after the cutoff].” From this phrasing, a reader might think that the *model* changes when we evaluated pre-cutoff problems vs. post-cutoff problems, but in fact we kept the model exactly the same for all problems. Now we understand that your actual meaning was that (for example) the evaluation procedure might have changed between pre- and post-cutoff data, or other subtle changes.
> >
> > With respect to the evaluation process itself, first, we note that from our side, everything remains the same pre- and post-cutoff: we keep the LLM exactly the same, and we use the same slightly modified version of [DeepMind’s evaluation handle](https://github.com/google-deepmind/code_contests) for all problems. From Codeforces’ side, the expected answer format is unchanged from early problems to late problems. Codeforces problems expect submissions of the following format: the submission must comprise a complete code file that, when executed, reads some data in from stdin and produces results in stdout (note: occasionally, there are problems with an alternative “interactive” evaluation strategy in which the submitted program can provide temporary responses that lead to more input being provided to the program; these problems have always been omitted from our analysis).
> > We therefore believe formatting details to be ruled out as an explanation for the shift in performance. There could potentially be more subtle changes, but we are not able to see any from looking at the problem statements and LLM outputs associated with the randomly selected subset of CodeForces problems released before and after cutoff (for varying levels of GPT functional correctness) in Appendix B.8.1.
> >
> > Finally, in fact, we have just finished adding initial [Code Bison](https://cloud.google.com/vertex-ai/docs/generative-ai/code/code-models-overview) (Google’s code generation foundation model, released around the same time as PaLM 2) results for Codeforces to our manuscript. Interestingly, we find a statistically significant shift in behavior before Code Bison’s cutoff of February 2023, and we do *not* find a significant shift in behavior if we artificially set the cutoff to Sept 2021 (GPT’s cutoff). This result gives further evidence to rule out “silly possibilities” such as a shift in Codeforces formatting, since we otherwise wouldn’t see such big changes for two different models, at their two different respective cutoff dates.
> >
> > Thank you once again for your comments; we hope that we have clarified our first response, and we are glad that our paper now has more evidence to rule out extraneous factors. Please let us know if you have any further questions or comments. We would be happy to reply any time before the author response period closes tomorrow (end of day Nov 22nd)!

---

### Official Review · Reviewer_XbnJ · 2023-11-01

**Soundness:** 3 good
**Presentation:** 3 good
**Contribution:** 3 good
**Rating:** 6
**Confidence:** 2

**Summary:**

This paper presents a detailed investigation into data contamination in large language models (LLMs), using GPT model training cutoffs to analyze benchmarks released over time. It examines two datasets, Codeforces and Project Euler, revealing clear patterns that suggest contamination based on the LLMs' pass rates correlated with benchmarks' GitHub popularity and release dates. The authors provide a comprehensive dataset, findings, and a framework for future analysis, promoting better practices for benchmark releases in the era of web-scale LLM training.

**Strengths:**

The idea to investigate data contamination in LLMs via cutoff datasets makes sense and is interesting, which guarantees that the testing data are not available in the training set of LLMs. And the findings are surprising, revealing that people should deal with the ability of LLMs more carefully. This study shows that LLMs are likely to have generalization problems as well as traditional ML models and deep neural networks. And I think this should raise the attention of ML researchers.

**Weaknesses:**

I am not quite familiar with LLMs, and I only have one question about the design of cutoffs. What if a code problem released later is exactly similar as some problems that has already existed? And how to measure the data contamination problem is also important.

**Questions:**

Please refer to Weaknesses.

---

> ### Author Response · Authors · 2023-11-16
> **Reply to Reviewer XbnJ (1/2)**
>
> We thank you for your thoughtful review. We appreciate that you find our longitudinal study interesting and that it should be raised to the attention of ML researchers.
>
> **1. Design of the cutoff: similar code problems.**
> Thank you for the great question, and we understand that your question lies in the methodology of our analysis.
>
> Recall that we propose a method to measure contamination through a natural experiment where the training cutoff bifurcates an evaluation set. We appreciate that you agree with this methodological approach and our resulting analysis, and we agree that checking for duplicated questions before and after the cutoff is an important step.
>
> In summary, based on your suggestion, we did find 56 exact duplicates, none of which straddle the cutoff (0.7% of the total set of problems). Community experts identified 8 similar questions in the Codeforces problems; only 5 straddled pre- and post-cutoff. The overall effect on our results should be minimal considering the large drop-off in performance after the cutoff. In the coming days, we will rerun our analysis and update you and the paper before the end of the rebuttal period. We explain more below.
>
> For *textual duplicates*, we have confirmed that there are none in Project Euler. In Codeforces, we have identified a set of 40 unique problems (out of a total of more than 8,300) for which the dataset contains multiple (i.e., 2-3) copies. This can occur if a competition problem is later re-posted as part of a practice set. All duplicated tuples belong to the subset of problems released before the GPT training cut-off. As the focus of our analysis is on comparing performance on examples released before versus after the training cutoff, we would be concerned if a majority of these examples were “cut-crossing”, but do not find that to be the case. Before the discussion period concludes, we will re-run our analyses omitting the duplicates, and will provide you with updated results. We expect that the impact of removing 56 observations (corresponding to the duplicates of the aforementioned 40 problems) from the pre-cutoff subset which contains >6,000 observations will be minimal, and do not expect the qualitative nature of our conclusions to change.
>
> As for *semantic duplicates*, it is certainly non-trivial to check for the existence of such problems. Thankfully, the Codeforces community has [discussed](https://codeforces.com/blog/entry/113016) this topic before, highlighting 8 near-duplicate pairs, of which 5 were pairs with one question on either side of the cutoff, and the other three were pairs with both questions were released before the cutoff.
>
>
> For the sake of examining those 5 pairs which straddled the cutoff, we compare GPT-4 performance:
> Pre-Cutoff Problem | Post-Cutoff Problem | Pre-Cutoff Problem Pass Rate | Post-Cutoff Problem Pass Rate
> |-|-|-|-|
> 765_F | 1793_F | 0.50 | 1.00
> 652_C | 1771_B | 0.38 | 0.00
> 342_E | 1790_F | 1.00 | 0.50
> 923_B | 1795_C | 0.17 | 0.00
> 1462_C | 1714_C | 0.00 | 0.00
> Mean before cutoff: 0.41, Mean after cutoff: 0.30
>
> We also compare GPT-3.5-Turbo performance:
> Pre-Cutoff Problem | Post-Cutoff Problem | Pre-Cutoff Problem Pass Rate | Post-Cutoff Problem Pass Rate
> |-|-|-|-|
> 765_F | 1793_F | 0.50 | 1.00
> 652_C | 1771_B | 0.00 | 0.00
> 342_E | 1790_F | 0.00 | 0.50
> 923_B | 1795_C | 0.33 | 0.00
> 1462_C | 1714_C | 0.14 | 0.00
> Mean before cutoff: 0.20, Mean after cutoff: 0.30
>
> A potential risk of including such semantically similar problem instances in our evaluation is that the performance on post-cutoff examples would be upwardly biased, since the model is actually seeing examples which are quite similar to those it has seen in training, despite our best efforts to control for such exposure via our cutoff-based partition of the dataset. The small size of this subset means this effect, were it to exist, would be relatively minimal, and it would be biased against, rather than in favor of, the conclusions we ultimately draw. As such, we do not feel that the presence of such examples undermines our analysis. We will provide an updated analysis in the coming days to confirm.
>
> Finally, we’d like to acknowledge that we agree with your broader point -- i.e., the question of *how* to measure or detect when data contamination has occurred is important. We contend that our approach, which leverages the cutoff date as a source of naturally arising variation, should be viewed as a compelling complement to existing approaches, which often rely on the ability to manipulate the underlying dataset and/or synthetically introduce contamination and then measure the impact on downstream tasks. We are able to determine that contamination is likely to have occurred by evaluating performance leveraging a freely available feature of the dataset (i.e., each problem’s release date, relative to the cutoff), without requiring any sort of post-facto intervention or manipulation.
>
>
>
> (1/2)

---

> ### Author Response · Authors · 2023-11-16
> **Reply to Reviewer XbnJ (2/2)**
>
> **2. Additional experimental updates**
> Finally, we would like to highlight a few additional experiments and updates we finished during the rebuttal period:
> 1. We have rendered the coefficient tables as forest plots (suggested by EfiM) and supplemented the regression tables with them throughout. See Figures 2 and 3 in the main paper, and associated figures in the Appendix. This change is very helpful for the reader to quickly visualize the regression coefficients and see how after the coefficients become closer to 1 (or have no effect) after the cutoff.
> 2. We have conducted new analyses to assess whether the drop-off in performance that we observe for problem examples released after the GPT training cutoff might be attributable to (potentially latent) covariate shifts.
> 3. We added a new section B.6 to describe our experiments with open source LLMs; we also have plans to expand to other LLMs during the rebuttal period and are awaiting results.
> 4. We added examples of generations from the LLMs in Section B.8.
>
>
> Thank you once again, for your excellent questions. Please let us know if you have any follow-up questions or further suggestions. We would be happy to continue the discussion, and we will update you with our reanalysis soon.
>
> (2/2)

---

> > ### Author Response · Authors · 2023-11-22
> > **Follow up to Reviewer XbnJ**
> >
> > We would like to highlight for you that in our last pdf revision, we updated our functional correctness regression analyses with the removal of the duplicate Codeforces problems. You can see in Figures 10-25 and Tables 1-9 that the removal did indeed have *no impact* on the conclusions of our work. We have also included other updates to our paper which significantly strengthen it, and you can see a summary of them [here](https://openreview.net/forum?id=m2NVG4Htxs&noteId=mQVSo5ovS7). Thank you again for your work to help improve our paper and your thoughtful review!

---

> > > ### Comment · Reviewer_XbnJ · 2023-11-23
> > > **Thank you**
> > >
> > > Thanks for your rebuttal, and most of my concerns are addressed. Since I'm not an expert in this field, I would like to maintain my score.

---

### Official Review · Reviewer_VA65 · 2023-11-05

**Soundness:** 1 poor
**Presentation:** 3 good
**Contribution:** 2 fair
**Rating:** 5
**Confidence:** 2

**Summary:**

The paper conducted longitudinal analysis of data contamination in large language models (LLMs), a problem where models are evaluated using data that they may have been trained on, thus overstating their capabilities.  The authors leveraged natural experiments provided by the training cutoff dates of models like GPT-3.5 and GPT-4 to study contamination. They analyzed Codeforces and Project Euler, websites that release code problems over time, and find evidence of contamination based on the pass rate of LLMs for problems released before their training cutoff dates. The study demonstrates statistically significant associations between a problem’s presence on GitHub and LLM performance for pre-cutoff problems.

**Strengths:**

1: The analysis from longitudinal perspective is novel.
2: The comprehensive experiments, large-scale dataset and code base provided by this work will definitely benefit the community of contamination analysis.
3: This paper is well organized and easy to understand.

**Weaknesses:**

1: The results are interesting but not that surprising. Many blogs or discussion in the community about Data Contamination has involved similar results.
2: There is lack of depth analysis about how implicit contamination is possible. If some real examples can be extracted to show how this could happen, it will be much better.

Overall, I do appreciate the effort to investigate the Data Contamination problem from longitudinal side and open-source data/codes. The experiments also show intriguing results. But I believe the contribution of this paper is not enough to be accepted by ICLR, for its limited scope and technical novelty. It's limited to Code datasets. And the only novelty is how to split the "train" and "test" set.

**Questions:**

N/A

---

> ### Author Response · Authors · 2023-11-16
> **Reply to Reviewer VA65 (1/2)**
>
> We thank you for taking time to review the manuscript and provide valuable feedback. We are glad to see that you find our work to be beneficial to the community of contamination analysis, our results intriguing, and our paper well-written. We reply to each of your questions below.
>
> **1. results interesting but not surprising.**
> Thank you, we agree that the results are interesting. While we understand that the results may not be surprising to you, they are to other reviewers; XbnJ described “the findings are surprising, revealing that people should deal with the ability of LLMs more carefully”. More importantly, our work is a non-trivial contribution to the community that presents the first scientifically rigorous confirmation of the phenomena that social media posts have speculated about via small-scale/ad hoc analyses. While many informal statements have been made about GPT-4 memorization, we show the first statistically significant differences in performance on problems released before and after the cutoff date. Furthermore, our work lays the groundwork for rigorous analysis of contamination and best practices in the age of LLMs trained on webscale data. Our methodology is becoming increasingly relevant as the community acknowledges the limitations of static benchmarks for LLM evaluation, and shifts toward dynamic/longitudinal benchmarks such as the ones we construct, open-source, and analyze here.
>
> **2. On how implicit contamination is possible.**
> To begin, we’d like to clarify our intention for including the words “implicit” and “explicit” in our abstract, specifically in reference to how a given problem might come to be included in a model’s training corpus. Our intention here was to acknowledge that the process of collecting webscale data can lead to the inclusion (in the training set) of examples that were never *intentionally* selected for training–for example, we consider the unintentional inclusion of BIG-bench examples in the GPT-4 training corpus ([OpenAI, 2023](https://arxiv.org/abs/2303.08774)). Through re-posting, removal of watermark information, indirect scraping, or other means, information intended to be omitted from LLM scraping is rarely truly safe. For clarity, we have modified our abstract text to refer to examples that are “intentionally” (explicitly) or “unintentionally” (implicitly) included in the training data.
>
> On a separate note, we have newly added Appendix B.8 which includes a number of samples of generated outputs from GPT-4 and GPT-3.5-Turbo. These can be used to qualitatively examine their outputs across a range of pass rates on both datasets.
>
> **3. On scope and novelty.**
> Thank you for your comments. We focus on code generation for a few key reasons: (i) it is a very popular use-case; (ii) most recent LLMs have code as a major part of their training data; (iii) unlike general online natural language text, GitHub has a uniform interface and is easily scrapable and cleanable, making it simultaneously easy for GitHub solutions to be added to train datasets as well as for us to assess GitHub presence; and (iv) code generation and problem solving datasets have objective correctness metrics (test cases) producing objective evaluations of open-ended generations that don’t require another model or human in the loop.
>
> We also note that dataset desiderata for the methodological approach we propose include: (i) problems must have been released over a sufficiently long time-horizon, such that it is possible to partition examples into pre- and post- GPT training cutoff subsets based on problem release date.  This restrictions precludes some other popular benchmarks which have been released in a single time-step (e.g. [HumanEval](https://github.com/openai/human-eval) or [MBPP](https://arxiv.org/pdf/2108.07732.pdf)). (ii) problems should consist of high-quality questions requiring non-trivial solution generation that admit objective evaluation functions/correctness measures.
> We have thus focused our efforts on the non-trivial scraping and processing required to analyze Project Euler and Codeforces datasets in the way that we have. We also emphasize that our work introduces tools and paves the way for further rigorous study related to data contamination, which gives our work the potential to have high impact in the community.
>
> We’d also like to highlight that our paper should also be viewed as a methodological one. Specifically, we believe our work takes a novel view on data contamination estimation by employing a natural experiment – a concept borrowed from the economic literature. We use this concept effectively here and argue for its further use, particularly in experimenting, evaluating, and discovering the intricacies of LLMs. We argue that this evaluation methodology should be used in particular on closed-source models which refuse to reveal critical and important details about their development/training. We have updated our manuscript to highlight this point further.
>
> (1/2)

---

> ### Author Response · Authors · 2023-11-16
> **Reply to Reviewer VA65 (2/2)**
>
> Thank you very much, once again, for your helpful feedback. If you find our responses satisfying, we respectfully ask that you consider increasing your score. We are still conducting new analyses and are excited to update you with their results. We would be very happy to answer any follow-up or additional questions you have.
>
> (2/2)

---

### Author Response · Authors · 2023-11-22
**Response to all Reviewers**

We thank all the reviewers for their insightful feedback and suggestions. We present the first thorough, longitudinal analysis of LLM data contamination, showing that LLMs’ ability to solve coding problems changes dramatically as a function of metrics such as problem release date and GitHub popularity. We appreciate that reviewers find that our work is **well-motivated** and an **important issue** for the community (VA65, XbnJ, EfiM, yfjE). Furthermore, we appreciate that most reviewers found our **experiments well-executed** (VA65, EfiM, yfjE) and our paper **well-written** (VA65, EfiM). We have now updated our paper to include all reviewer suggestions. We highlight a few of the main changes below:
* **We included analysis for two new models: Google’s [Code Bison](https://ai.google/discover/foundation-models/) (released around the same time as  PaLM 2) and Code-Llama 34b Instruct.** Code Bison achieves strong performance on Codeforces, and we are able to show the exact same statistical significance of the differing pass rates before and after its assumed training cutoff date of February 2023. In addition to complementing our results, this gives ample evidence that our conclusions are not based on extraneous factors such as a subtle difference in the evaluation protocol before and after the cutoff date. On Code-Llama 34b Instruct, we were not able to get Code-Llama to output answers which yielded pass rates above 1% on Codeforces. We qualitatively observe that many outputs are very far away from a correct answer, for example, giving code in C++ or non-English text despite English prompting asking for Python code. We will shortly add the raw results to our anonymous repository.
* We provide evidence that several possible covariate shifts are not responsible for the observed drop off in performance in problems released after the GPT cutoff.
* We added 32 examples of pairs of Codeforce problems and GPT outputs, sorted into categories based on LLM model, release date, and functional correctness. These examples allow readers to see qualitative features of the output code, and they act as an additional check to rule out additional extraneous factors.

Thank you once again, for your reviews, and we would be happy to answer any follow-up questions.

---

### Meta-Review · Area_Chair_2bKM · 2023-12-18

**Metareview:**

Reviewers were largely positive about this work, which demonstrates clear evidence for data contamination via in coding and math problems based on prevalence of related information on GitHub.  The authors accomplish this by exploiting discontinuities in training data. This is of broad interest to the community and provides a simple mechanism for evaluating leakage.  A few pieces of feedback for the CR.

1. The visualizations of the regression models such as the marginal effects plots in Fig1 or Fig12 provide a more interpretable explanation of the results of the regression analysis than reporting the logistic regression coefficients and reporting on the log-odds ratios.  I would recommend using such visualizations or reporting simple differences in means when describing the average effect of the cutoff period.
2. It would be nice to see or have some clear reporting on the average change in performance before/after on the benchmarks, rather than seeing the means broken out by prevalence.
3. log(difficulty) itself is not inherently meaningful to the reader, and I am not sure how these values were chosen.  I would recommend reporting on the quartiles or some other percentile-based breakdown.  Having visualizations of the distribution of difficulties (e.g., as a histogram, density, CDF) would also be informative for the SM.
4.  What do the investigations in the authors' submission tell us about the prevalence of misleading results in recent literature?  For example, how well do the problems reported in the work reflect benchmarks used in the literature?  The GPT-4 whitepaper appears to indicate that GPT-4 fails to complete 50% of the codeforce tests, and the authors' analysis includes some control for contamination (with seemingly identical results for both).  In this latter case, why is there such a discrepancy here? While the details of the contamination testing protocol are not well defined in the GPT-4 paper, perhaps the authors can share additional details via personal correspondence or some updated version of the arxiv paper.
5. The fact that code-llama performs so poorly on codeforce is quite surprising.  Is it possible that there is a bug in the evaluation?  In any case some more discussion here would be nice to see.
6. Saying that the authors take an "experimental economics view" is a little bit of an overstatement.  Indeed, while this is an example of a discontinuity that would delight many econometricians, the analysis is neither experimental (you could call it a quasi-experiment) nor specific to the type of identification strategy commonly used in the social sciences, statistics, and epidemiology community.
7. This is perhaps beyond the scope of this paper, but seeing similar results on more diverse tasks could help readers understand the generality of the results.  This in turn gives the community a stronger sense of how prevalent this issue is, and give LLM researchers more ideas for how to reduce contamination.

**Justification For Why Not Higher Score:**

The analysis is not particularly extensive and could consider more diverse benchmark problems.  This limits the authors' ability to speak to the prevalence of these types of issues in the literature (or how general the risk is).

**Justification For Why Not Lower Score:**

This work is of broad interest and represents (as far as I am aware) the first rigorous study of this kind.  My confidence is a little low here since I am not an expert in dataset leakage.

---

### Decision · Program_Chairs · 2024-01-16

Accept (poster)